# Interactive Speculative Planning: Enhance Agent Efficiency through Co-design of System and User Interface

Wenyue Hua[*,1], Mengting Wan[2], Shashank Vadrevu[2], Ryan Nadel[2],
Yongfeng Zhang[1], and Chi Wang[†,3]

[1]Rutgers University, New Brunswick, [2]Microsoft, [3]Google Deepmind

## Abstract

Agents, as user-centric tools, are increasingly deployed for human task delegation, assisting with a broad spectrum of requests by generating thoughts, engaging with user proxies, and producing action plans. However, agents based on large language models (LLMs) often face substantial planning latency due to two primary factors: the efficiency limitations of the underlying LLMs due to their large size and high demand, and the structural complexity of the agents due to the extensive generation of intermediate thoughts to produce the final output. Given that inefficiency in service provision can undermine the value of automation for users, this paper presents a human-centered efficient agent planning method – Interactive Speculative Planning – aiming at enhancing the efficiency of agent planning through both system design and human-AI interaction. Our approach advocates for the co-design of the agent system and user interface, underscoring the importance of an agent system that can fluidly manage user interactions and interruptions. By integrating human interruptions as a fundamental component of the system, we not only make it more user-centric but also expedite the entire process by leveraging human-in-the-loop interactions to provide accurate intermediate steps.

## 1 Introduction

Large language models (LLMs) have demonstrated strong reasoning abilities (Zhang et al., 2024c; Qiao et al., 2022; Fan et al., 2023; 2024; Jin et al., 2024), enabling them to plan and interact with external tools and the real world. This has led to the development of LLM-based agents, which have become popular as task solvers and human assistants. Various agent frameworks have been created to facilitate these applications, including single-agent systems such as Langchain (Topsakal & Akinci, 2023), OpenAGI (Ge et al., 2024), and HuggingGPT (Shen et al., 2024), as well as multi-agent systems like AutoGen (Wu et al., 2023), MetaGPT (Hong et al., 2023), BabyAGI (Nakajima, 2023), and Camel (Li et al., 2023). Numerous methods have also been proposed to enhance the performance of LLM-based agents, ranging from chain-of-thought (Wei et al., 2022), tree-of-thought (Yao et al., 2024), ReAct (Yao et al., 2022), Reflexion (Shinn et al., 2024), to multi-agent discussion (Chan et al., 2023) systems.

These high-performing advancements in agents often come at the expense of **time efficiency** (Zhou et al., 2024; Ding et al., 2024b; Zhang et al., 2024b), which can be attributed to two main factors: (1) the underlying backbone language model can be inefficient due to its growing size and high request volume, and (2) the complex agent structure, such as tree-of-thought and ReAct, requires generating prolonged thought before the final answer, leading to extended waiting time and increased token generation costs, and blue (3) the sequential nature of action steps in plans, where one action must be completed before the next can begin, also exacerbates the situation. However, not all steps in agent

---

[*]Work done as Microsoft Research intern.
[†]Work done while working at Microsoft Research.

planning necessitate computationally intensive thought processes, making the universal application of complex agent architectures or agents with advanced backbone LLMs inefficient.

Moreover, as demonstrated in classic human-computer interaction studies (MacKenzie & Ware, 1993), system latency is a critical factor in shaping user experience. Prolonged delays during human-computer interaction can lead to increased stress levels, frustration, decreased task performance, and reduced user satisfaction (MacKenzie & Ware, 1993; Jota et al., 2013; Barron et al., 2004; Simpson et al., 2007; Carr et al., 1992). While LLM-based agent systems are designed to assist users, few have been designed to prioritize user experience, which is arguably one of the most important aspects of any human-AI interactive systems. Particularly in scenarios where complex tasks are delegated to LLM-based agents, often involving high stakes and complex decision-making processes, users may not anxiously wait for the agent to respond all at once, but rather expect the agent to provide timely feedback so that they can effectively direct the process (Lubars & Tan, 2019; Hemmer et al., 2023). Thus we argue that a fully automated "blackbox" agent system with prolonged response delays is suboptimal for user experience—a prevalent issue in most LLM-based agent systems today. Instead, a responsive time-efficient mixed-initiative system is required to address this challenge (Horvitz, 1999). Thus we focus on designing a synchronous user interaction where a human is in the loop and waiting for agent response in real time.

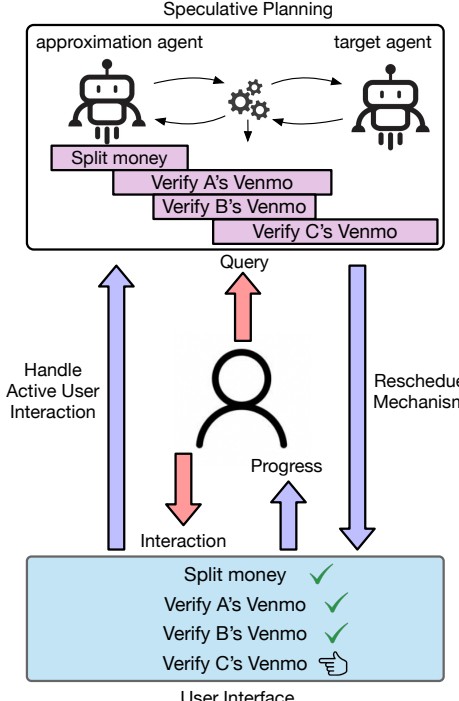

**Figure 1:** Interactive Speculative Planning: user query is handled by speculative planning with approximation and target agent. Then a rescheduling mechanism serializes the computed result on UI and enables the user to actively interact with the system for further acceleration. Finger-pointed action is the action intervened by the user.

This work aims to address the time efficiency and perceived latency issue from both algorithmic system design and human interaction perspectives. We introduce an interactive efficient planning algorithm, **Interactive Speculative Planning**, marking the first LLM-based agent system dedicated to enhancing time efficiency as well as managing human interactions and interruptions. As demonstrated in Figure 1, this approach seamlessly integrates time efficiency and human-in-the-loop interaction, anticipating and managing user engagement during periods of extended latency. By treating user input as intermediate results, the system accelerates the overall process, thereby improving both system time efficiency and user experience. Consequently, this system offers a more user-centric and time-efficient solution for agents as human delegates.

**System-Level Parallelization** The system-level algorithm, speculative planning, is inspired by recent advancements in speculative decoding (Leviathan et al., 2023; Liu et al., 2023a; Chen et al., 2023; Spector & Re, 2023; Liu et al., 2024; Cai et al., 2024). It leverages a dual-agent framework: an efficient yet less capable *approximation agent*, and a slower but more powerful *target agent*. For each task, the approximation agent generates action steps sequentially. Concurrently, for every step produced by the approximation agent, the target agent is asynchronously invoked to generate the subsequent step, using the current trajectory from the approximation agent as a provisional prefix. In this setup, calls to the approximation agent are *sequential*, whereas those to the target agent are *asynchronous*. For each action step, if the outputs of both agents match, the process continues; otherwise, the approximation agent is halted, and its output is replaced by the target agent's output to maintain performance integrity. This strategy potentially reduces the time a target agent takes to complete the task to that of the approximation agent, thereby enhancing time efficiency. It should

be noted that while we consider a single user interface of the agent system, the system backend can be built with various architectures, including a multi-agent design.

**UI-Level Rescheduling** Note that under the speculative planning algorithm, target agent calls are asynchronous, leading to non-sequential outputs. To facilitate user interaction, we introduce a UI-level rescheduling algorithm (Oh et al., 2024; Cheng et al., 2024; Mei et al., 2024; Jawahar et al., 2023; Srivatsa et al., 2024) that presents both the approximation and target agent's results sequentially. This sequential presentation enables users to accurately perceive the computational latency imposed by the target agent. Consequently, they may intervene at their discretion, such as when a computation step is prolonged or yields erroneous results. Therefore, Interactive Speculative Planning enables active human involvement in the decision-making process, allowing users to interrupt extended processes and evaluate whether to accept or rectify the algorithm's outputs. This human-in-the-loop approach complements the algorithmic evaluation of the approximation agent's results, making the system more user-centric.

In summary, with active user intervention, Interactive Speculative Planning can be viewed as an interactive framework involving **three** agents: the approximation agent, the target agent, and the human agent. These three agents collaborate and interleave their operations to collectively accelerate the overall agent planning process.

## 2  RELATED WORK

Various agent systems (Xi et al., 2023; Liu et al., 2023b; Ge et al., 2023) have been developed including single agent such as Huginggpt (Shen et al., 2024), OpenAGI (Ge et al., 2024), and BabyAGI (Nakajima, 2023), and multi-agent systems (Du et al., 2023) such as AutoGen (Wu et al., 2023; Zhang et al., 2024a) and Camel (Li et al., 2023), based on the strong reasoning ability (Wu et al., 2024; Zhang et al., 2023b) and common sense knowledge (Kwon et al., 2024) encoded in LLMs. To improve the performance of LLM-based agents, various methods have been proposed. The most basic approach is the chain-of-thought (Wei et al., 2022), where the LLM generates a step-by-step thought process for each action. More advanced methods include ReAct (Yao et al., 2022), where the agent thinks before acting, and Reflexion (Shinn et al., 2024), where the agent thinks, acts, and reflects on its decisions. The tree-of-thoughts (Yao et al., 2024) method involves the agent thinking several steps ahead before acting. Additionally, multi-agent discussion systems (Du et al., 2023; Hua et al., 2023; Lin et al., 2024; Wu et al., 2023) have been developed, where multiple agents discuss and debate to enhance performance. It is generally observed that stronger backbone models and more complex multi-LLM interaction usually leads to better performing agents (Wang et al., 2024; Li et al., 2024; Chen et al., 2024).

However, these improvements in agent performance often come at the expense of time efficiency as longer thought processes result in extended waiting times. Although agent task can be intricate, and sometimes only the most powerful models may be capable of executing them effectively as suggested by (Xie et al., 2024), not all steps within a task are equally challenging to plan and generate (Zhang et al., 2023a; Saha et al., 2024). Therefore, a dynamic selection of appropriate LLMs for specific tasks can be a viable strategy to balance performance and efficiency/cost.

Numerous methods have been developed to enhance agent efficiency (Zhang et al., 2023a; Ding et al., 2024a; Saha et al., 2024). EcoAssistant (Zhang et al., 2023a) is the first system aimed at cost-efficient agents, initiating tasks with the most economical agent and switching to more capable and expensive agents only upon failure of the cheaper alternative. The System-1.x Planner (Saha et al., 2024) introduced a controllable planning framework using language models, capable of generating hybrid plans and balancing between complex and simple agent planning strategies based on problem difficulty, potentially offering both time and cost efficiency. However, the System-1.x Planner is limited to specific planning strategies and necessitates extensive training. In contrast, our proposed Interactive Speculative Planning can adopt any combination of approximation and target agent in a training-free manner, guaranteeing performance that is at least equivalent to, and potentially superior to (with user interventions), that of the target agent alone.

## 3  INTERACTIVE SPECULATIVE PLANNING

Interactive Speculative Planning is a collaborative framework that enhances the efficiency and accuracy of agent planning by integrating the efforts of three agents: the approximation agent, the target

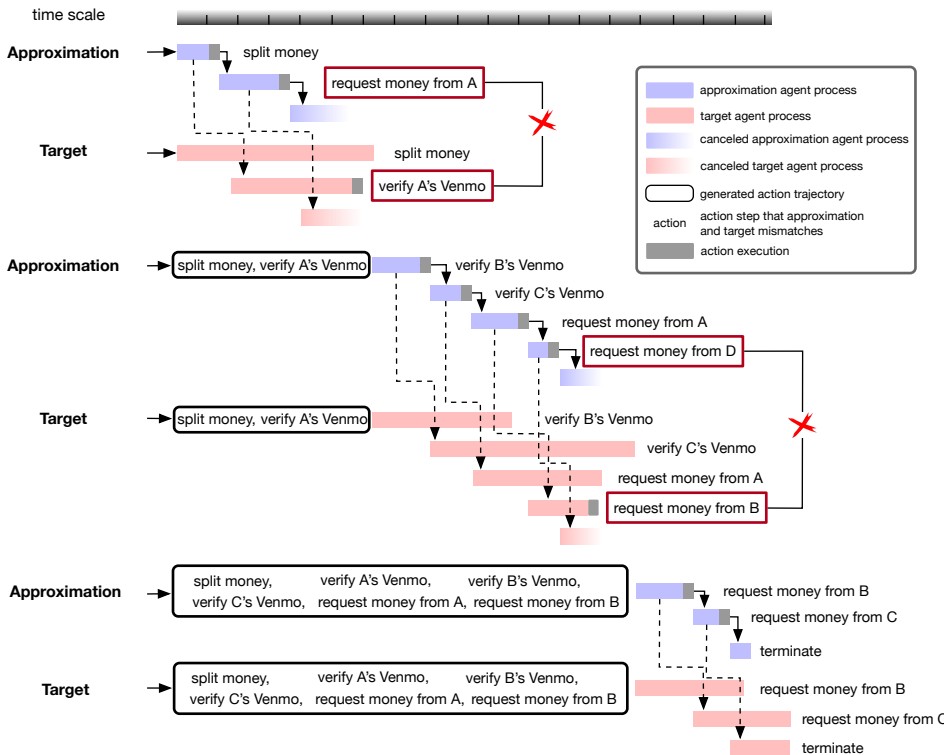

**Figure 2:** A demonstration of the Speculative Planning Algorithm, where the cross symbol indicates the step where the $\mathcal{A}$'s computed result differs from that of $\mathcal{T}$.

agent, and the human agent. The approximation agent generates quick but potentially inaccurate steps, while the target agent verifies and refines these steps. The human agent intervenes to correct or optimize the speed of planning process, ensuring that the final plan aligns with user expectations. This interactive approach accelerates the overall planning process and enhances user experience by reducing response delays and allowing for real-time adjustments.

In this section, we introduce the system-level speculative planning algorithm, provide a detailed analysis of its time efficiency, and then discuss the anticipated user interaction paradigm and how it is incorporated into the system.

**Speculative Planning Algorithm** is a technique designed to accelerate the planning process of a slow but powerful target agent $\mathcal{T}$. It utilizes a fast and efficient approximation agent $\mathcal{A}$ to draft the plan step by step. Instead of waiting for $\mathcal{T}$ to finish all preceding steps, $\mathcal{A}$'s action history is used as a prefix for $\mathcal{T}$ to generate the next step. This approach allows $\mathcal{T}$ to begin processing subsequent steps sooner, effectively speeding up the overall planning process by leveraging the groundwork laid by $\mathcal{A}$. For every length-$i$ prefix (the first $i$ steps) generated by $\mathcal{A}$, both agents generate the $(i+1)$-th step simultaneously, without waiting for $\mathcal{T}$ to finish the $i$-th step. If the $i$-th step of the plan generated by both agents matches after $\mathcal{T}$ finishes it, then the more efficient but less capable agent $\mathcal{A}$ is deemed to have correctly computed the step, and $\mathcal{T}$'s $(i+1)$-th step computed based on it is usable. Time is thus saved because the time for $\mathcal{T}$ to compute steps $i$ and $i+1$ is reduced to the time taken by $\mathcal{A}$ to compute step $i$ and $\mathcal{T}$ to compute step $i+1$. However, if there is a mismatch, it implies that $\mathcal{A}$ has erred at the $i$-th step, and we will replace $i$-th step with $\mathcal{T}$'s result. Furthermore, all concurrent calls of $\mathcal{A}$ and $\mathcal{T}$ with prefixes longer than $i$ must be halted and discarded, as they are based on an incorrect prefix and their results are unusable. In short, this algorithm achieves time savings by having $\mathcal{T}$ utilize the result generated by the fast $\mathcal{A}$ as a prefix to generate the next step, rather than waiting for prefix steps from the slower $\mathcal{T}$ to be completed.

Figure 2 illustrates a scenario where a person and their friends A, B, and C go to a restaurant, pay the bill, and need to split the cost among themselves. In this example, the time to compute the first two steps "Split money, verify A's Venmo" using speculative planning is shorter than using normal

| $n$ | the number of planning steps for a task |
|---|---|
| $time(\mathcal{A}, s)$ | the time the approximation agent $\mathcal{A}$ takes to generate step $s$ in the plan |
| $time(\mathcal{T}, s)$ | the time the target agent $\mathcal{T}$ takes to generate step $s$ in the plan |
| $e(s)$ | the time to execute a step $s$ in the plan and return an observation |
| $start\_time(\mathcal{T}, s_i)$ | it equals to $\Sigma_{j=b+1}^{j=i-1}(time(\mathcal{A}, s_j) + e(s_j))$ which indicates the start time of $\mathcal{T}$ to generate step $s_i$ since the one previous breaking point $b$. Notice that $start\_time(\mathcal{T}, s_i) = start\_time(\mathcal{A}, s_i)$ |
| $end\_time(\mathcal{T}, s_i)$ | it equals to $\Sigma_{j=b+1}^{j=i-1}(time(\mathcal{A}, s_j) + e(s_j)) + time(\mathcal{T}, s_i)$, which indicates the end time of $\mathcal{T}$ of generate step $s_i$ since the one previous breaking point $b$ |

**Table 1:** Notation Summary

agent planning: Speculative Planning requires the sum of $\mathcal{A}$'s time to compute the first step and $\mathcal{T}$'s time to compute the second step; In contrast, normal agent planning requires the sum of $\mathcal{T}$'s time to compute the first step and $\mathcal{T}$'s time to compute the second step. However, as the second step computed by $\mathcal{A}$ – "request money from A" does not match $\mathcal{T}$'s output – "verify A's Venmo", all following steps based on the action trajectory that includes $\mathcal{A}$'s incorrect second step become invalid. This includes the third process of $\mathcal{T}$, the third step from $\mathcal{A}$, and so on.

To prevent an excessive number of concurrent target agent processes, we introduce a hyperparameter $k$ to the speculative planning algorithm. This parameter sets a limit on the maximum number of steps that $\mathcal{A}$ that can sequentially generate and being executed before all corresponding target agent processes are completed. By controlling the value of $k$, users can flexibly manage the maximum number of concurrent target agent processes. Speculative planning algorithm is presented in Algorithm 1 in Appendix A.

**Time Efficiency Analysis** Here we analyze the time efficiency improvement brought by the speculative planning algorithm. We summarize the notations in Table 1. When we do not utilize speculative planning, the time taken to generate and execute the whole plan is $\Sigma_{i \le n}(time(\mathcal{T}, s_i) + e(s_i))$. To compute the time when employing speculative planning, we first define the list of breaking steps $B$, which consists of indices $i$ of steps $s$ in the plan where the sequential generation of $\mathcal{A}$ is halted, *i.e.* when $\mathcal{A}$'s prediction $a_i = \mathcal{A}(i)$ differs from $\mathcal{T}$'s prediction $t_i = \mathcal{T}(i)$ for the $i$-th step in the planning, as well as when the number of consecutive speculative steps generated by the approximation process reaches the hyperparameter $k$[1].

The time taken to generate and execute the entire plan is then determined by the following equation, where the time to compute each sequence of steps between two consecutive elements $B_i$ and $B_{i+1}$ in $B$, which is determined by step $i$ that takes longest time to compute for the target agent $\mathcal{T}$:

$$\Sigma_{B_i \in B[:-1]}(\max\{(end\_time(\mathcal{T}, s_j) \mid B_i + 1 \le j \le B_{i+1}\}) \tag{1}$$

Best case scenario is that no step generated by $\mathcal{A}$ differs from the step generated by $\mathcal{T}$. Thus in this specific case, the breaking points $B$ is simply all numbers $i$ smaller than $n$ that are divisible by $k$, and the computing time for the best case is:

$$\Sigma_{i \in \{i \bmod k = 0 \mid i < n\}}(\max_{i \le j < i+k} end\_time(\mathcal{T}, s_j))) \tag{2}$$

Worst case scenario is that all steps generated by $\mathcal{A}$ are rejected by $\mathcal{T}$. In this extreme case, the set of breaking steps, $B$ comprises all integers from 0 to $n - 1$. Under these circumstances, the time taken to generate and execute the plan downgrades to normal agent planning, as no steps from $\mathcal{A}$ is usable and no time can be saved. Assuming there is no latency to compare $\mathcal{A}$'s result and $\mathcal{T}$'s result, the total time is simply calculating the sum of the time taken to generate and execute each step in the plan by $\mathcal{T}$ sequentially:

$$\Sigma_{0 \le i \le n-1}(time(\mathcal{T}, s_i) + e(s_i)) \tag{3}$$

**The aforementioned worst-case scenario demonstrates that, in terms of time efficiency, speculative planning is upper-bounded by the time taken in non-speculative planning.** More analysis

---

[1]For notational convenience, we add $-1$ as the first element and $n - 1$ as the last element in the list $B$

on different aspects of speculative planning including total token generated which correspond to the cost and maximum number of concurrent LLM/API calls required are presented in Appendix C. The impact of different settings on time efficiency including speed and accuracy of $\mathcal{A}$, the choice of $k$ and their interleaved relationship are presented in Appendix C.4.

**User Interface and Interaction Design**    Now we present the interaction component of Interactive Speculative Planning. The user interface (UI) has two main objectives: (1) from the aspect of perception, it aims to provide the user with an easy-to-follow result and a basic understanding of the algorithm's inner workings, allowing the user to see $\mathcal{T}$'s computation time for each step and how $\mathcal{A}$ is saving time; (2) from the aspect of interaction, it aims to provide system support for the user to actively interact with or interrupt the ongoing agent processes – when $\mathcal{T}$ is taking too long for a step or neither $\mathcal{A}$ nor $\mathcal{T}$ provides a satisfying step proposal during generation. Therefore, the user interface, together with the underlying system mechanism design, primarily addresses two key aspects: (1) what the users should see and (2) how the system can handle user interactions.

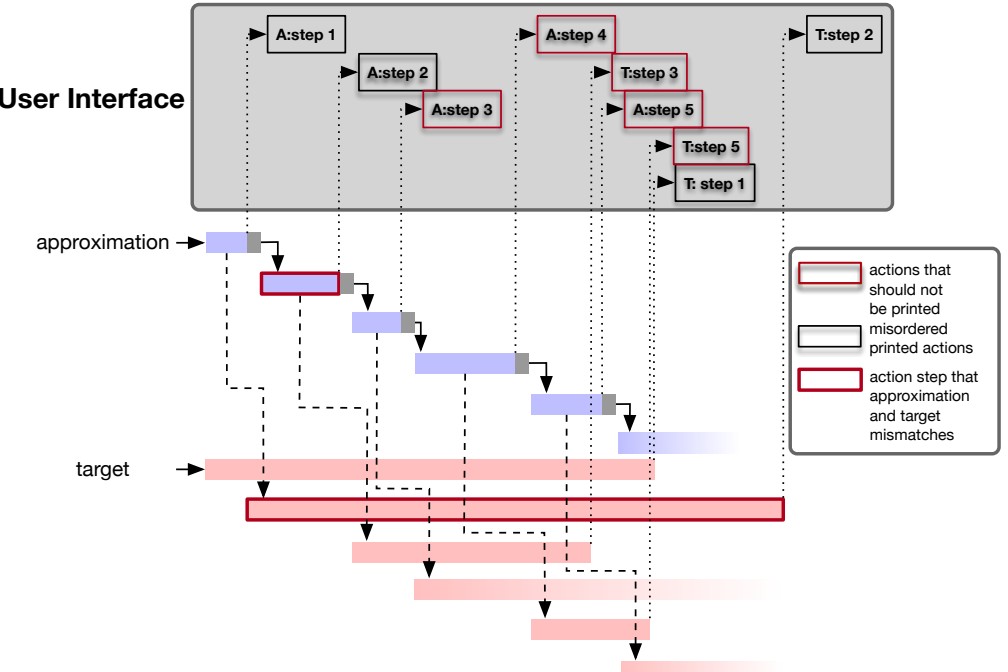

**Figure 3:** User interface issues stemming from immediate presentation of computed action steps.

For the first goal, notice that immediately printing the outputs of $\mathcal{A}$ and $\mathcal{T}$ upon generation can be very confusing for two reasons: (1) some outputs of $\mathcal{A}$ and $\mathcal{T}$ should not be shown to the user at all, and (2) the outputs of $\mathcal{T}$ are misordered. Figure 3 presents an example scenario for the two issues. For issue 1: $\mathcal{A}$'s output on the second step of the plan mismatches with $\mathcal{T}$'s output, and thus all results generated by $\mathcal{A}$ based on the mistaken "step 2" will ultimately be discarded. However, an immediate output of the agent's generation will present $\mathcal{A}$'s computed steps "step 3, 4, 5" and $\mathcal{T}$'s computed steps "step 3, 4" which are generated based on the wrong prefix. For issue 2: as all $\mathcal{T}$'s calls are asynchronous, the time for each step to finish will not follow a sequential order, and thus an immediate printing out of the generated output will not be sequential either. Therefore, a rescheduling mechanism is needed to provide a clear presentation of the algorithm.

To ensure an understandable user interface to track the agents' progress and facilitate user intervention, the presented output is rescheduled by a Reschedule Mechanism. This mechanism allows the user to view verified and to-be-verified computed steps of $\mathcal{A}$ and $\mathcal{T}$ with minimal perceived latency. The Reschedule Mechanism, shown in Algorithm 2 in Appendix B, takes the queue of $\mathcal{A}$ processes and the queue of $\mathcal{T}$ processes as input, tracing the last printed out message from either $\mathcal{A}$ and $\mathcal{T}$, and then decide which message to present next to the user: (1) it presents the $i$-th step from $\mathcal{A}$ only after all preceding steps from $\mathcal{A}$ have been confirmed to be consistent with $\mathcal{T}$, ensuring that no steps computed based on unverified prefixes are presented (2) it presents the $i$-th step from $\mathcal{T}$ only after all preceding steps from $\mathcal{T}$ have been presented, ensuring a sequential order. This design not only

ensures a sequential presentation but also highlights the time difference between $\mathcal{A}$ and $\mathcal{T}$, allowing the user to identify which action is bottlenecking the program.

For the second goal, we enable users to *actively* interrupt the program at any time. Unlike current user interface designs in various agent systems (Wu et al., 2023) where users are allowed to interact with the system when being *passively* prompted to input information or opinions, we believe that users are more inclined to actively engage in the agent task delegation process (Lubars & Tan, 2019). We handle user interaction in two common scenarios: (1) when noticing excessive perceived latency between the last presented output of $\mathcal{A}$ and the next output of $\mathcal{T}$ (assuming $\mathcal{A}$'s generation speed is sufficiently fast that users would not typically interrupt it), and (2) when dissatisfied with the outputs of $\mathcal{T}$ for a given step. For the first scenario, since the user interface presentation for the $i$-th step of the plan can indicate the latency $l_i$ between the presentation of the $i$-th approximation output and the $i$-th target output $t_i$, users can choose to interrupt during the time of $l_i$ and input their own value. The underlying system will handle this keyboard interruption by halting the $i$-th process of $\mathcal{T}$, incorporating the user's input into the agent action trajectory without disturbing other concurrent processes. In the second scenario, users are able to interrupt the program if they deem the results from $\mathcal{T}$ unsatisfactory for a given step. During the brief presentation of the output $t_i$ for any step $i$, users can intervene and input their preferred optimal step for step $i$ as an oracle. The algorithm 2 is presented in Appendix B. The experiment to study the impact of user interruption and time efficiency is provided in Appendix C.4.

## 4 EXPERIMENT

This section presents the results on two agent planning benchmarks: OpenAGI (Ge et al., 2024) and TravelPlanner validation (Xie et al., 2024). For each benchmark, we implement speculative planning with four different configurations.

**Benchmarks** OpenAGI is a benchmark designed for agent planning with complex tasks, built on computer vision and natural language processing-related tasks. Tools accessible to the agents include "Sentiment Analysis", "Machine Translation", "Object Detection", "Visual Question Answering," etc. An example task in the benchmark is "Restore noisy, low-resolution, blurry, and grayscale images to regular images," whose solution is a sequence of tool usage: "Image Super-resolution, Image Denoising, Image Deblurring, Colorization." This benchmark contains 117 multi-step tasks.

TravelPlanner is a benchmark focusing on travel planning. It provides a rich sandbox environment, various tools for accessing nearly four million data records, and meticulously curated planning intents and reference plans. These plans also involve many constraints, including budget constraints, environmental constraints, etc. An example task is "Please plan a travel itinerary for me. I'm departing from Cincinnati and heading to Norfolk for three days. The dates of travel are from March 10th to March 12th, 2022. I have a budget of $1,400 for this trip." whose solution contains a sequence of actions such as "FlightSearch[Cincinnati, Norfolk, 2023-03-12]" where "FlightSearch" is the function name for the action while "Cincinnati, Norfolk, 2023-03-12" is the natural language free-form parameter for "FlightSearch". We experiment on the validation dataset containing 180 datapoints.

**Speculative Planning Settings** To experiment with Interactive Speculative Planning in real-life scenarios, we demonstrate the performance by four different combinations of $\mathcal{A}$ and $\mathcal{T}$ as settings containing the most commonly used agent architecture as $\mathcal{T}$.

**Setting 1** $\mathcal{A}$ employs direct-generation-based planning with a GPT-4-turbo backbone, while $\mathcal{T}$ utilizes ReAct-based planning (Yao et al., 2022) with the same backbone. For each step in the plan, $\mathcal{T}$ uses ReAct to first deliberate on the action and then generate it through two separate API calls, whereas $\mathcal{A}$ directly generates the action for that step.
**Setting 2** $\mathcal{A}$ uses direct-generation-based planning with a GPT-4-turbo backbone, and $\mathcal{T}$ employs chain-of-thought (CoT)-based planning with the same backbone. For each step in the plan, $\mathcal{T}$ uses CoT to first reason and then generate the result in a single API call, while $\mathcal{A}$ directly generates the action for that step.
**Setting 3** $\mathcal{A}$ uses CoT-based planning with a GPT-4-turbo backbone, and $\mathcal{T}$ system uses multi-agent-debate (MAD) including 2 agents with 2 rounds of discussion on every step of the plan with a GPT-4-turbo backbone. For each step in the plan, $\mathcal{T}$ system has two agents discuss with each other and finalize the action to take for the current step, and the while $\mathcal{A}$ uses CoT to first reason and then generate the result in a single API call for the step.

**Setting 4** $\mathcal{A}$ uses direct-generation-based planning (DG) with a GPT-3.5-turbo backbone, and $\mathcal{T}$ uses direct-generation-based planning with a GPT-4-turbo backbone. In this setting, both $\mathcal{A}$ and $\mathcal{T}$ directly generate the result for each step. Notice that we cannot provide results for TravelPlanner in this setting, as direction generation using GPT-3.5-turbo fail to provide a valid action in many cases.

In all experiments, we set $k = 4$. We utilized one OpenAI API for experiments under Settings 1, 2, and 4, and two OpenAI APIs (one API for each agent in the multi-agent system) for experiments under Setting 3. For the OpenAGI benchmark, given its limited action space, we used *exact match* to verify the correctness of the output generated by $\mathcal{A}$ against the output of $\mathcal{T}$. This ensured that the output of the speculative planning is the same as that of normal agent planning. For the TravelPlanner benchmark, which contains a much larger action space, each action is a combination of a function name from a fixed set and natural language free-form parameters. We verified the consistency between the output of $\mathcal{A}$ and the output of $\mathcal{T}$ based on an exact match of the function name and a soft match of the natural language parameters. The soft match is implemented by computing the Levenshtein distance: if the function name matched and the Levenshtein distance is smaller than 0.3, then the action is verified. As we leverage soft match to verify the output of $\mathcal{A}$, it is not guaranteed that the result from speculative planning remains the same as the result from normal agent planning and therefore we also provide the accuracy performance of both normal agent planning and speculative planning, presented in Appendix D.

| Metrics | Settings | | | | | | | |
|---|---|---|---|---|---|---|---|---|
| | Setting 1 | ReAct | Setting 2 | CoT | Setting 3 | MAD | Setting 4 | DG |
| TT | $33.91_{\pm30.38}$ | $43.63_{\pm25.39}$ | $28.64_{\pm25.49}$ | $39.96_{\pm27.25}$ | $105.42_{\pm50.84}$ | $182.70_{\pm421.49}$ | $4.63_{\pm1.78}$ | $5.77_{\pm1.83}$ |
| Min-TT | 6.80 | 9.16 | 3.53 | 8.60 | 28.24 | 50.89 | 1.70 | 2.23 |
| ST | $5.92_{\pm3.00}$ | $8.69_{\pm2.75}$ | $5.52_{\pm3.71}$ | $7.98_{\pm2.72}$ | $21.50_{\pm6.69}$ | $34.84_{\pm58.94}$ | $1.14_{\pm0.25}$ | $1.49_{\pm0.43}$ |
| Min-ST | 2.33 | 4.41 | 0.50 | 3.81 | 11.70 | 19.21 | 0.75 | 1.03 |
| TO | $1920_{\pm879.79}$ | $1812.89_{\pm832.30}$ | $1770.61_{\pm1010.44}$ | $1397.90_{\pm794.55}$ | $6781.43_{\pm3159.84}$ | $4075.4_{\pm1603.54}$ | $107.05_{\pm38.76}$ | $40.13_{\pm13.39}$ |
| Min-TO | 760 | 652 | 455 | 352 | 1754 | 1441 | 47 | 17 |
| SO | $288.72_{\pm65.29}$ | $266_{\pm44.37}$ | $281.92_{\pm88.77}$ | $229.45_{\pm44.23}$ | $1385_{\pm391.77}$ | $836.65_{\pm112.06}$ | $26.47_{\pm5.06}$ | $10.14_{\pm1.98}$ |
| Min-SO | 190.00 | 166.58 | 143.83 | 162.8 | 877 | 558.33 | 19.25 | 8.5 |
| MC | $4.66_{\pm0.59}$ | $1_{\pm0.00}$ | $4.49_{\pm0.82}$ | $1_{\pm0.00}$ | $4.53_{\pm0.56}$ | $1_{\pm0.00}$ | $4.05_{\pm0.21}$ | $1_{\pm0.00}$ |
| Min-MC | 3 | 1 | 3 | 1 | 3 | 1 | 4 | 1 |
| cost | $\$0.122_{\pm0.072}$ | $\$0.0713_{\pm0.026}$ | $\$0.074_{\pm0.040}$ | $\$0.044_{\pm0.018}$ | $\$0.2973_{\pm0.1387}$ | $\$0.2160_{\pm0.0795}$ | $\$0.0012_{\pm0.0011}$ | $\$0.0012_{\pm0.0004}$ |

**Table 2:** Main experiment results on OpenAGI benchmark.

**Evaluation Metrics** In terms of time efficiency, we report the cross-dataset average and minimum total generation time (recorded in second), as well as the cross-dataset average and minimum stepwise generation time, for all planning tasks across experimental setting, compared with the normal planning setting. It is important to note that the total generation time heavily depends on the number of steps in the plan, which can be influenced by randomness. Therefore, we provide the stepwise generation time, which mitigates the effect of randomness related to the number of steps. To provide a comprehensive understanding of the algorithm, we also include metrics related to the total number of tokens generated during the process and the total API cost.

In total, there are 11 metrics: (1) total time (TT), (2) the minimum total time across the dataset (min-TT), (3) stepwise time (ST), (4) the minimum stepwise time across the dataset (min-ST), (5) Total tokens generated (TO), (6) the minimum total tokens generated across the dataset (min-TO), (7) stepwise tokens generated (SO), (8) the minimum stepwise tokens generated across the dataset (min-SO), (9) maximum concurrent API calls (MC) (10) the minimum maximum concurrent API calls across the dataset (min-MC), (11) the average total cost used to finish the plan (cost).

## 4.1 MAIN EXPERIMENT RESULT

Table 2 presents the results on OpenAGI dataset. In Setting 1 where $\mathcal{T}$ utilizes ReAct, we cut the total running time on average by 22.27% percentage and the stepwise running time on average by about 31.87%. In Setting 2 where $\mathcal{T}$ utilizes CoT, we cut the total running time on average by 28.32% and the stepwise running time on average by about 30.83%. In Setting 4 where both $\mathcal{A}$ and $\mathcal{T}$ uses direct generation but with different backbone models, we cut the total running time on average about 20.37% percentage and the stepwise running time on average by about 23.50%. *Setting 3 includes a very slow $\mathcal{T}$ using a multi-agent debate; we obtain the largest efficiency improvement: this setting can cut the total time by 42.30% and the stepwise running time on average by 38.29%.* But notice that though time efficiency is enhance, there is an increase at cost: in the first and second settings, there is an 70% increase in monetary cost; in the third setting, the computation time decreased by about 40%, at the cost of a 37% increase in monetary cost which is relatively small because multi-agent discussion is itself a very expensive prompting method; in the last setting, where

we use GPT-3.5 as the approximation agent, the computation time decreased by about 20% at almost no additional cost due to the very cheap nature of GPT-3.5. Thus, we can see that the increase in cost can be mitigated if $\mathcal{A}$'s prompting method is simpler or $\mathcal{A}$'s backbone model is cheaper.

| Metrics | Settings | | | | | | | |
| --- | --- | --- | --- | --- | --- | --- | --- | --- |
| | Setting 1 | ReAct | Setting 2 | CoT | Setting 3 | MAD | Setting 4 | DG |
| TT | $137.33_{\pm66.39}$ | $176.28_{\pm77.18}$ | $98.09_{\pm45.02}$ | $121.37_{\pm32.18}$ | $568.10_{\pm292.99}$ | $733.12_{\pm290.51}$ | - | - |
| Min-TT | 40.78 | 55.18 | 29.22 | 42.09 | 149.00 | 127.59 | - | - |
| ST | $11.16_{\pm5.49}$ | $14.13_{\pm3.61}$ | $10.71_{\pm5.50}$ | $12.75_{\pm4.33}$ | $27.53_{\pm8.67}$ | $40.03_{\pm8.74}$ | - | - |
| Min-ST | 4.53 | 7.04 | 2.65 | 4.92 | 12.33 | 23.06 | - | - |
| TO | $3751.94_{\pm853.86}$ | $2460.95_{\pm332.07}$ | $3082_{\pm235.09}$ | $2002.93_{\pm276.54}$ | $12353.84_{\pm5872.86}$ | $8976.39_{\pm5371.31}$ | - | - |
| Min-TO | 1389 | 1762 | 833 | 1329 | 3443 | 2049 | - | - |
| SO | $298.84_{\pm128.97}$ | $246.13_{\pm56.34}$ | $220.79_{\pm56.19}$ | $197.08_{\pm87.68}$ | $733.18_{\pm477.72}$ | $591.65_{\pm467.82}$ | - | - |
| Min-SO | 128.13 | 108.30 | 85.42 | 68.06 | 189.00 | 186.27 | - | - |
| MC | $5_{\pm0.00}$ | $1_{\pm0.00}$ | $5_{\pm0.00}$ | $1_{\pm0.00}$ | $5.00_{\pm0.00}$ | $1_{\pm0.00}$ | - | - |
| Min-MC | 5 | 1 | 5 | 1 | 5 | 1 | - | - |
| cost | $\$0.1583_{\pm0.0367}$ | $\$0.1038_{\pm0.0033}$ | $\$0.1393_{\pm0.0241}$ | $\$0.0874_{\pm0.0125}$ | $\$0.5941_{\pm0.2871}$ | $\$0.3990_{\pm0.2309}$ | - | - |

**Table 3:** Main experiment results on TravelPlanner benchmark.

Table 3 presents the results on TravelPlanner validation dataset. Similar to the experiment on OpenAGI dataset, we can find noticeable time efficiency improvement when using speculative planning: in Setting 1, the average latency on total generation time has decreased for 21.43% while the stepwise generation time has decreased for 29.52%; Setting 2 has decreased average total time by 19.18% and the stepwise generation time by 32.53%; Setting 3 has decreased average total time by 25.46% and the stepwise generation time by 31.69%. Similar increase of cost can be observed as in the dataset OpenAGI: all the three settings observe a 50% increase in cost.

## 4.2 ANALYSIS OF TIME IMPROVEMENT BREAKDOWN

Having observed the average time efficiency improvement for the two datasets across the four settings, we provide a more granular analysis of the time efficiency improvement based on the quality of $\mathcal{A}$'s output across datapoints in the datasets. Specifically, we aim to examine how much time is saved given a specific level of $\mathcal{A}$'s accuracy. This analysis will allow us to identify the sources of time savings and determine which datapoints, at which levels of accuracy, contribute to time efficiency improvement. For demonstration, for each dataset and each setting, we present two figures: one displaying the distribution of datapoints with different levels of accuracy and the other displaying the average stepwise time efficiency improvement proportion for all levels of accuracy.

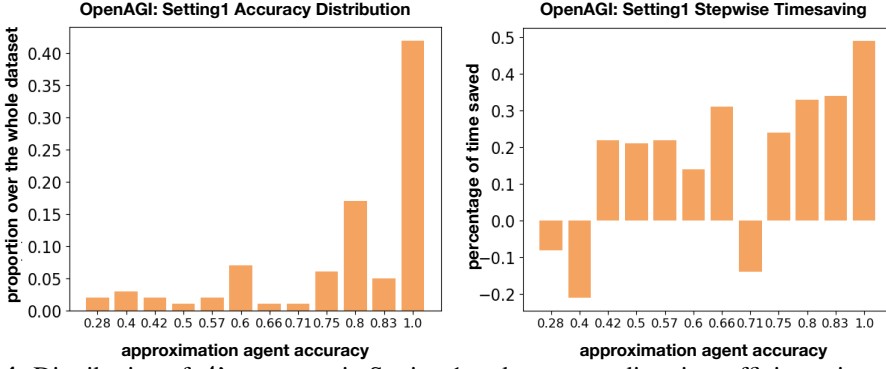

**Figure 4:** Distribution of $\mathcal{A}$'s accuracy in Setting 1 and corresponding time efficiency improvement

Figure 4, 5, 7 and 6 demonstrate the breaking-down results on OpenAGI dataset. Notably, Setting 1, 2, and 3 contain a significant proportion of datapoints that exhibit a perfect accuracy of $\mathcal{A}$ which correspond to the largest time efficiency improvement. Nevertheless, for datapoints with lower accuracy, a substantial reduction in stepwise generation time can also be observed. This trend is consistent across all settings. And notice that in Setting 3, stepwise time saving proportion can achieve almost 60% when $\mathcal{A}$'s accuracy achieves higher then 80%. Analysis for TravelPlanner dataset is presented in Appendix E.

However, it is important to note that in almost all settings for both datasets, there are data points exhibiting "negative" time efficiency improvement, i.e., longer stepwise running times when applying speculative planning compared to normal agent planning. Most of these cases occur when

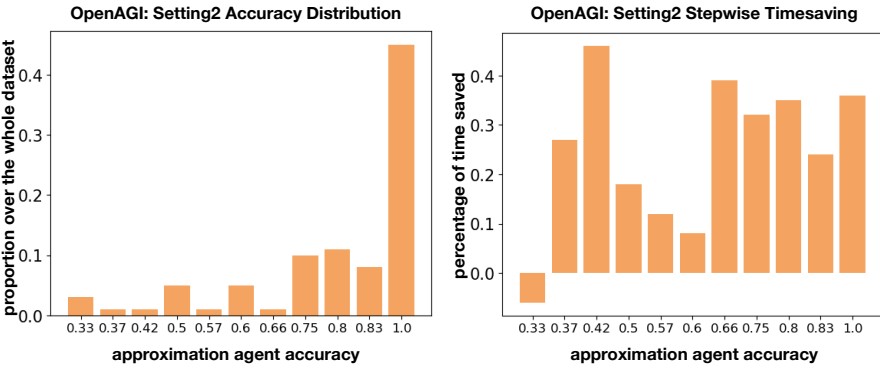

**Figure 5:** Distribution of $\mathcal{A}$'s accuracy in Setting 2 and corresponding time efficiency improvement

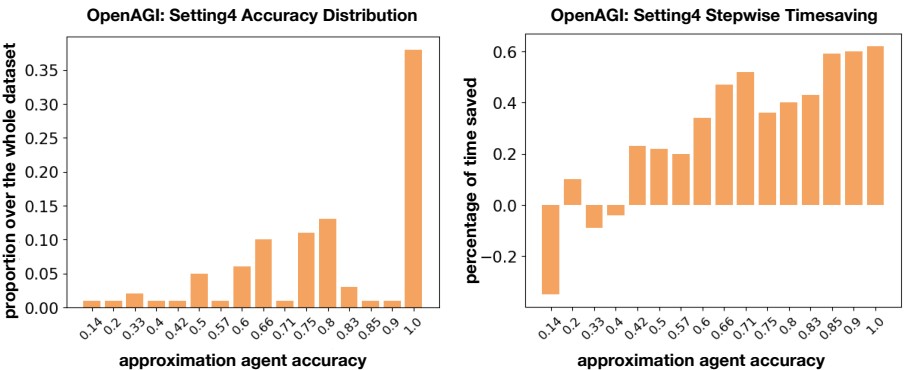

**Figure 6:** Distribution of $\mathcal{A}$'s accuracy in Setting 3 and corresponding time efficiency improvement

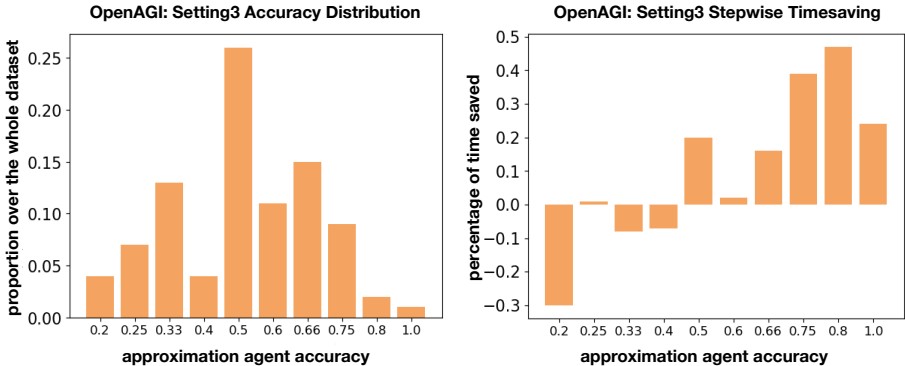

**Figure 7:** Distribution of $\mathcal{A}$'s accuracy in Setting 4 and corresponding time efficiency improvement

the accuracy of $\mathcal{A}$ is relatively low, a scenario in which latency efficiency analysis suggests limited time efficiency improvement but no worse than normal agent planning, contrary to what we observe. This discrepancy can be attributed to two assumptions in the theoretical analysis: (1) The speeds of $\mathcal{A}$ and $\mathcal{T}$ are constant across different runs on the same data points. However, in actual usage, there is significant randomness involved due to the number of tokens generated in each step, causing variations in the speed of both $\mathcal{A}$ and $\mathcal{T}$ even for the same data point. (2) The speeds of $\mathcal{A}$ and $\mathcal{T}$ are not affected by multiple concurrent queries. In practice, $\mathcal{T}$ runs in parallel, meaning the API for $\mathcal{T}$ must process multiple concurrent queries, which may also slow down the overall speed for each individual call of $\mathcal{T}$.

## 5 CONCLUSION

This paper introduces Interactive Speculative Planning, a novel approach that integrates an efficient agent system with an active user interface to enhance and accelerate agent planning with human interaction. By treating human interruptions as an integral part of the system, we not only make the planning process more user-centric but also accelerate the entire system by providing correct intermediate steps. More analysis and limitations are presented in Appendix F and G.

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

# A   SPECULATIVE PLANNING ALGORITHM

---

**Algorithm 1:** Speculative Planning Algorithm.

---

**Input:** Approximation agent: $\mathcal{A}$, Target agent: $\mathcal{T}$, task prompt $p$, action trajectory $\mathcal{S} = [\,]$,
      TEMINATE=False, max approximation steps $k$

**Output:** $\mathcal{S}$

1  i = 0

2  **while** *not* TEMINATE **do**

3     approximation_step = 0

4     **for** *approximation_step ≤ k* **do**

5        **DoParallel**

6           create async process $\text{APPROXIMATION}_i = \mathcal{A}(p, \mathcal{S})$
                         `// will return i-th action step `$a_i$` by running `$\mathcal{A}$

7           approximation_step += 1

8           create async process $\text{TARGET}_i = \mathcal{T}(p, \mathcal{S})$
                         `// will return i-th action step `$t_i$` by running `$\mathcal{T}$

9       $a_i$ = await $\text{APPROXIMATION}_i$
                      `// wait for `$\mathcal{A}$` to finish computation sequentially`

10      $o_i = \text{EXECUTION}(a_i)$
             `// execute the generated plan step `$a_i$` and obtain observation `$o_i$

11     update $p$ by adding description about $a_i$ and $o_i$

12     i += 1

13     $\mathcal{S}$.append($[a_i, o_i]$)
               `// cache generated step `$a_i$` and corresponding observation `$o_i$

14     **for** *j = 0 to i* **do**

15        **if** $t_j$ *is computed* & $t_j \neq a_j$ **then**

16           $o'_j = \text{EXECUTION}(t_j)$
                     `// re-execute the generated plan step `$t_j$` and obtain`
                 `observation `$o'_i$

17        $\mathcal{S} = \mathcal{S}[:j] + [[t_j, o'_j]]$
                                     `// update cache`

18          update $p$ by modifying $i$-th step based on description about $t_j$ and $o'_i$

19          cancel all ongoing APPROXIMATION processes and $\text{TARGET}_k$ if $k > j$
                              `// cancel useless processes`

20          break the outer for loop and go to line 4

21     **if** $\mathcal{S}[-1][0]$ *is "terminate"* **then**

22        TEMINATE=True

23        break the outer for loop

24  **return** $\mathcal{S}$

---

# B    RESCHEDULING MECHANISM WITH USER INTERRUPTION

---

**Algorithm 2:** Rescheduling Mechanism with User Interruption.

---

1 **Function** `Register-Handler`(*target_tasks, target_task_id*) **:**
2    **Function** `Exit-Handler`(*signum, frame, target_tasks, target_task_id*) **:**
3       target_tasks[target_task_id].cancel()
4    **End Function**
5    signal.signal(signal.SIGTSTP, partial(`Exit-Handler`, target_tasks, target_task_id))
6    $t_{target\_task\_id}$ = user input for action step target_task_id
                      `// prompt user input to as oracle result`
7    $ts$.append($t_{target\_task\_id}$)
8 **End Function**
**Input:** Approximation process queue: $\mathcal{A}s$, Target process queue: $\mathcal{T}s$, Approximation result
        presentation index tracker $a\_tracker$, Target result presentation index tracker
        $t\_tracker$, Approximation result list $as$, Target result list $ts$
**Output:** $a\_tracker, t\_tracker, as, ts$
9 **if** $a\_tracker \leq t\_tracker$ **then**
10    $i = t\_tracker$
11    **if** *process* $\mathcal{A}s[i]$ *is completed* **then**
12       present $as[i]$ to user interface
13       Register-Handler(target_tasks=$\mathcal{T}s$, target_task_id=$t\_tracker$)
            `// setup signal handler to enable proper handling of user`
        `interruption when user is waiting for` $ts[i]$ `to be computed`
14       $a\_tracker$ += 1
15    **end**
16 **end**
17 **else**
18    $i = a\_tracker$
19    **if** *process* $\mathcal{T}s[i]$ *is completed* **then**
20       present $ts[i]$ to user interface but allow user to modify $ts[i]$ as $ts[i]'$
        `// enable user to directly change` $\mathcal{T}$`'s computed result` $ts[i]$ `after`
        `presenting it to user`
21       $t\_tracker$ += 1
22    **end**
23 **end**
24 **return** $a\_tracker, t\_tracker, as, ts$

---

# C    COMPLETE EFFICIENCY ANALYSIS

In this section, we will provide a theoretical analysis of the time savings (latency), total token generation requirement, and concurrent API call rate required by the speculative planning approach. Additionally, we will present simulated experiment results to support our analysis and demonstrate the effectiveness of the proposed method.

## C.1    TIME EFFICIENCY ANALYSIS

This subsection analyzes the time efficiency improvement brought by the speculative planning algorithm. We summarize the notations in Full Notation Summary 4.

When we do not utilize speculative planning, the time taken to generate and execute the whole plan is $\Sigma_{i \leq n}(time(\mathcal{T}, s_i) + e(s_i))$. To compute the time when employing speculative planning, we first define the list of breaking steps $B$, which consists of indices $i$ of steps $s$ in the plan where the sequential generation of $\mathcal{A}$ is halted, *i.e.* when $\mathcal{A}$'s prediction $a_i = \mathcal{A}(i)$ differs from $\mathcal{T}$'s prediction $t_i = \mathcal{T}(i)$ for the $i$-th step in the planning, as well as when the number of times where continuous speculative steps generated by the approximation process reaches the hyperparameter $k$. *For notational convenience, we include $-1$ as the first element and $n-1$ as the last element in $B$.*

| $n$ | the number of planning steps for a task |
| --- | --- |
| $time(\mathcal{A}, s)$ | the time the approximation agent $\mathcal{A}$ takes to generate step $s$ in the plan |
| $time(\mathcal{T}, s)$ | the time the target agent $\mathcal{T}$ takes to generate step $s$ in the plan |
| $e(s)$ | the time to execute a step $s$ in the plan and return an observation |
| $token(\mathcal{A}, s)$ | the token the approximation agent $A$ requires to generate step $s$ in the plan |
| $token(\mathcal{T}, s)$ | the token the target agent $T$ requires to generate step $s$ in the plan |
| $start\_time(\mathcal{A}, s_i)$ | it equals to $\Sigma_{j=b+1}^{j=i-1}(time(\mathcal{A}, s_j) + e(s_j))$, which indicates the start time of $\mathcal{A}$ to generate step $s_i$ since the one previous breaking point $b$ |
| $start\_time(\mathcal{T}, s_i)$ | it equals to $\Sigma_{j=b+1}^{j=i-1}(time(\mathcal{A}, s_j) + e(s_j))$ which indicates the start time of $\mathcal{T}$ to generate step $s_i$ since the one previous breaking point $b$. Notice that $start\_time(\mathcal{T}, s_i) = start\_time(\mathcal{A}, s_i)$ |
| $end\_time(\mathcal{A}, s_i)$ | it equals to $\Sigma_{j=b+1}^{j=i}(time(\mathcal{A}, s_j) + e(s_j))$, which indicates the end time of $\mathcal{A}$ of generate step $s_i$ since the one previous breaking point $b$ |
| $end\_time(\mathcal{T}, s_i)$ | it equals to $\Sigma_{j=b+1}^{j=i-1}(time(\mathcal{A}, s_j) + e(s_j)) + time(\mathcal{T}, s_i)$, which indicates the end time of $\mathcal{T}$ of generate step $s_i$ since the one previous breaking point $b$ |

**Table 4:** Full Notation Summary

The time taken to generate and execute the entire plan is then determined by the following equation, where the time to compute each sequence of steps between two consecutive elements $B_i$ and $B_{i+1}$ in $B$, which is determined by step $i$ that takes longest time to compute for the target agent $\mathcal{T}$:

$$\Sigma_{B_i \in B[:-1]}(\max\{(end\_time(\mathcal{T}, s_j) \mid B_i + 1 \le j \le B_{i+1}\}) \tag{4}$$

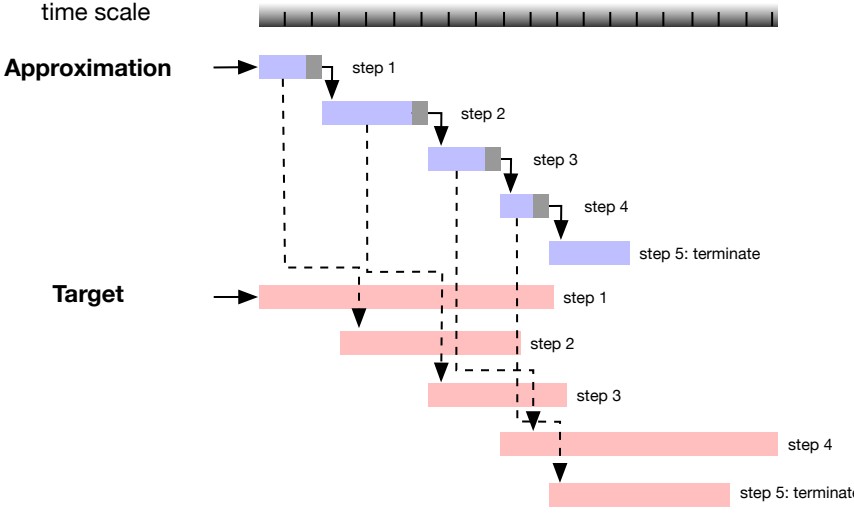

**Figure 8:** Best case scenario, assuming $k = n$

**Best case scenario** is that no step generated by $\mathcal{A}$ differs from the step generated by $\mathcal{T}$, as shown in Figure 8. Thus in this specific case, the breaking points $B$ is simply all numbers $i$ smaller than $n$ such that $i \bmod k = 0$, and the computing time for the best case is:

$$\Sigma_{i \in \{i \bmod k = 0 | i < n\}}(\max_{i \le j < i+k} end\_time(\mathcal{T}, s_j))) \tag{5}$$

**Worst case scenario** is that all steps generated by $\mathcal{A}$ are rejected by $\mathcal{A}$. a partial example is presented in Figure 9. In this extreme case, the set of breaking steps, $B$ comprises all integers from $0$ to $n - 1$.

Under these circumstances, the time taken to generate and execute the plan downgrades to normal agent planning. This equation calculates the sum of the time taken to generate and execute each step

in the plan sequentially, without any speculative planning. The total time can be expressed as:

$$\Sigma_{0 \le i \le n-1}(time(\mathcal{T}, s_i) + e(s_i)) \tag{6}$$

The aforementioned worst-case scenario demonstrates that, in terms of time efficiency, speculative planning is upper-bounded by the time taken in non-speculative planning. This implies that the maximum time required for speculative planning will not exceed the time taken by the traditional, non-speculative approach.

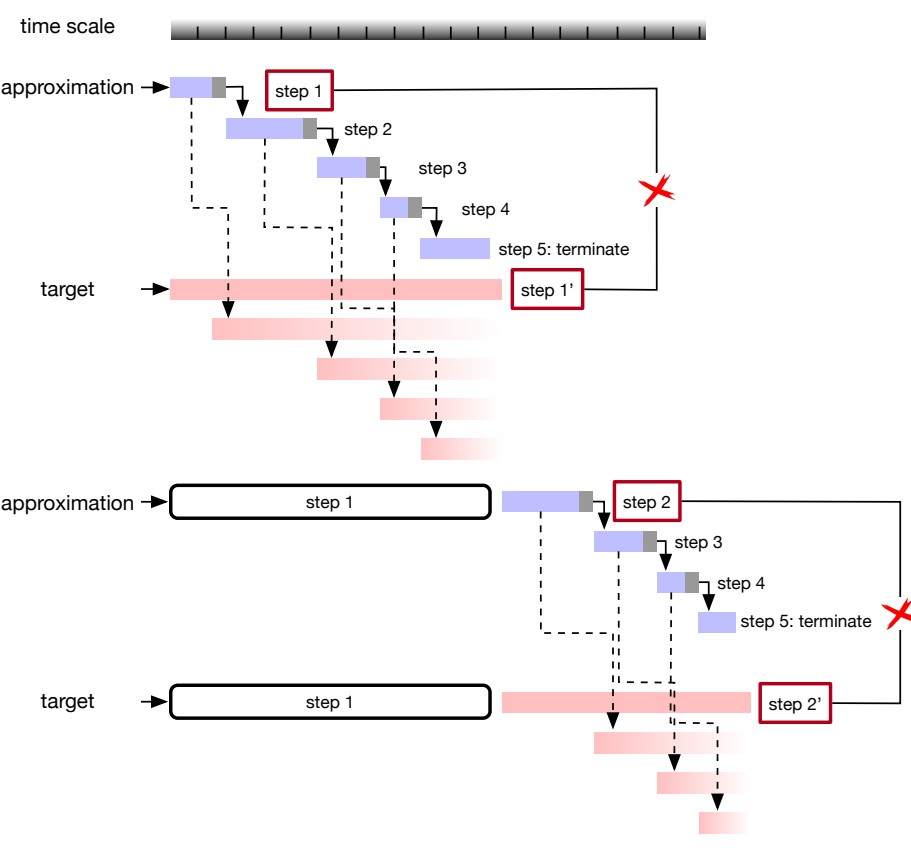

**Figure 9:** Worst case scenario

## C.2 TOTAL TOKEN REQUIRED

In this subsection, we analyze the total token generation when using the speculative planning algorithm.

When not utilizing speculative planning, the total number of tokens used to generate and execute the plan is $\Sigma_{i<n} token(\mathcal{T}, s_i)$. Speculative planning requires more tokens, as both $\mathcal{A}$ and $\mathcal{T}$ go through the entire plan at least once, potentially generating "wasted" tokens – proposed steps that are not used in the final plan which are computed based on incorrect prefix. Between any two breaking points $B_i$ and $B_{i+1}$, the number of tokens generated is the sum of tokens generated by $\mathcal{A}$ and $\mathcal{T}$ for steps $s_j$ in between as well as "unused/wasted" tokens generated by both $\mathcal{A}$ and $\mathcal{T}$ for any step $s_j$ such that $j \ge B_{i+1}$, where the process ends before $\mathcal{T}$ finishes all the process between any two breaking points $B_i$ and $B_{i+1}$. Thus, we represent the tokens generated $T_{B_i}$ between two consecutive breaking points $B_i$ and $B_{i+1}$ as below:

$$T_{B_i} = \underbrace{\Sigma_{j=B_i+1}^{j=B_{i+1}}(token(\mathcal{A}, s_j) + token(\mathcal{T}, s_j))}_{\text{sum of tokens generated by } \mathcal{A} \text{ and } \mathcal{T} \text{ in between } B_i \text{ and } B_{i+1}} + \underbrace{\Sigma_{j=B_{i+1}+1}^{M_i}(token(\mathcal{A}, s_j) + token(\mathcal{T}, s_j))}_{\text{wasted tokens}}$$

$$\tag{7}$$

where $Q = \max_{B_i < l \le B_{i+1}}\{end\_time(\mathcal{T}, s_l)\}$ is the ending time for all steps between $B_i$ and $B_{i+1}$ to be computed, and $M_i = \min\{\max\{l < n \mid end\_time(\mathcal{A}, s_l) \le Q\}, k + B_i\} - B_{i+1}$ is the number of wasted steps initiated by $\mathcal{A}$, that is, all processes that ends before $Q$ but are computed based on incorrect prefix.

Thus, the ultimate total number of tokens generated is the summation of $T_{B_i}$s:

$$\Sigma_{B_i \in B[:-1]} T_{B_i} \tag{8}$$

**Best case scenario** is that all steps generated by $\mathcal{A}$ matches those generated by $\mathcal{T}$, and therefore we do not have any "wasted" tokens, and then both $\mathcal{A}$ and $\mathcal{T}$ go through the agent generation plan. In this situation, $M_i = 0$ in the best case scenario for all $i$ corresponding to $B_i$ in $B$.

$$\Sigma_{0 \le i \le n-1}(token(\mathcal{A}, s_i) + token(\mathcal{T}, s_i)) \tag{9}$$

**Worst case scenario** is that none of the steps generated by $\mathcal{A}$ matches with those generated by $\mathcal{T}$. Additionally, each $\mathcal{T}$ process finishes after all $\mathcal{A}$ processes are completed, and the earliest called $\mathcal{T}$ process always finishes the last. Figure 9 represents a partial example. Formally, the worst case scenario will occur under the condition which can be expressed as:

$$\forall B_i \in B, end\_time(\mathcal{T}, s_{B_i+1}) \ge end\_time(\mathcal{A}, s_{B_i+1}) \text{ and} \tag{10}$$

$$\forall B_i < l \le B_{i+1}, end\_time(\mathcal{T}, s_{B_i+1}) \ge end\_time(\mathcal{T}, s_l) \text{ and} \tag{11}$$

$$\forall i \le n - 1, a_i \text{ does not match } t_i \tag{12}$$

In such a case, $Q = end\_time(B_i + 1)$ and $M_i = k - 1$ in the worst case scenario. Each $\mathcal{A}$ process $i$ will run for $(i \mod k) + 1$ times (for example, the first process where $i = 0$ runs for 1 time, the $k$-th process where $i = k - 1$ will run for $k$ times, and the $k + 1$-th process where $i = k$ will run for 1 time), and each $\mathcal{T}$ process $i$ is run for $i$ times. Consequently, the total number of tokens generated in this worst-case scenario is:

$$\Sigma_{i=0}^{n-1}((i \mod k) + 1) * (token(\mathcal{A}, s_i) + token(\mathcal{T}, s_i)) \tag{13}$$

## C.3 RATE REQUIRED

This subsection focuses on analyzing the rate required to run the speculative planning algorithm, which is determined by the maximum number of concurrently running agent calls.

When not utilizing speculative planning, all agent calls are executed sequentially. Consequently, the required rate, which is the maximum number of concurrently running agent calls, is 1. When using speculative planning, we naturally have at least 2 concurrent calls: 1 for $\mathcal{A}$ and 1 for $\mathcal{T}$. But it can be more than 2, as shown in Figure 2 where we can have many $\mathcal{T}$ processes running at the same moment. To determine the maximum concurrent $\mathcal{C}$ processes, we identify the target agent process that overlaps with the most other target processes and add 1 for the additional approximation process. For all $\mathcal{T}_l$ processes for $B_i < l \le B_{i+1}$, we find the $j$-th process $\mathcal{T}_j$ that overlaps with the most other $\mathcal{T}$ processes by:

$$\mathcal{T}_j = \max_{B_i < j \le B_{i+1}} \underbrace{|\{l < n \mid start\_time(\mathcal{T}, s_l) \le start\_time(\mathcal{T}, s_j) \le end\_time(\mathcal{T}, s_l)\}|}_{\text{count the number of target processes overlapping with process } j} \tag{14}$$

We denote the number of overlapping processes to be $C_{T_i}$. Notice that we have a hyperparameter $k$ set up which controls the number of sequential $\mathcal{A}$ calls can be conducted without waiting for all corresponding $\mathcal{T}$ calls to be finished. Therefore, Note that $C_{T_i}$ is upper-bounded by $k$. Since the concurrent processes are the overlapping target process plus the approximation process, $\mathcal{C}_{B_i} = C_{T_i} + 1$ which is upper-bounded by $k + 1$ between any consecutive $B_i$ and $B_{i+1}$.

Thus, the maximum concurrent $\mathcal{C}$ processes is the maximum of all $\mathcal{C}_{B_i}$:

$$\mathcal{C} = \max_{B_i \in B[:-1]} \mathcal{C}_{B_i} \tag{15}$$

**Best case scenario** is where there is exactly 2 concurrent processes running, 1 $\mathcal{A}$ process and 1 $\mathcal{T}$ process and there is no time overlap between any two $\mathcal{T}$ processes. This may only occur when for each step $s_i$, $time(\mathcal{T}, s_i) \le time(\mathcal{A}, s_i)$.

**Worst case scenario** is when there is a sequence of steps $i$ to $i + k$ such that $\forall i < j \le i + k, end\_time(\mathcal{T}, s_i) > start\_time(\mathcal{T}, s_j)$. In this case, there exists a time point where $k$ target processes are running concurrently, resulting in a total of $k + 1$ concurrent processes.

## C.4 SIMULATION EXPERIMENT FOR SPECULATIVE PLANNING

To elucidate the relationship between the performance of the Interactive Speculative Planning system and various hyperparameter configurations, we conducted three series of simulation experiments. Two experiments aimed to investigate the impact of different settings in speculative planning, specifically: (1) the choice of approximation agent $\mathcal{A}$, (2) the parameter $k$; and the third experiment investigates the impact of the number of user interruptions on overall latency. For the impact of $\mathcal{A}$, we examined $\mathcal{A}$'s accuracy relative to that of $\mathcal{T}$ (accuracy computed by treating $\mathcal{T}$'s result as ground truth), as well as $\mathcal{A}$'s computational speed. In the rest of the paper, we use $\mathcal{A}$'s accuracy to refer to the relative accuracy of $\mathcal{A}$ with respect to the result of $\mathcal{T}$.

For the simulation experiments, we set the following parameters unchanged: (1) the plan consists of 10 steps, (2) the generation speed of $\mathcal{T}$ is 8 seconds per action ($time(\mathcal{T}, s) = 8$) (3) for each step, $\mathcal{A}$ generates 10 tokens ($time(\mathcal{T}, s) = 10$) , (4) for each step, $\mathcal{T}$ generates 20 tokens ($time(\mathcal{T}, s) = 20$), and (5) for clarity, we set execution time to be 0 ($e(s) = 0$).

The first series of experiments explores the impact of $\mathcal{A}$'s accuracy with respect to $\mathcal{T}$ and the hyperparameter of $k$ planning time and total tokens generated. We fix the speed of $\mathcal{A}$ ($time(\mathcal{A}, s) = 2$) to be 2 seconds per action and vary $\mathcal{A}$'s accuracy in $\{0.0, 0.1, 0.2, 0.3, 0.4, 0.5, 0.6, 0.7, 0.8, 0.9, 1.0\}$ and the hyperparameter of $k$ in $\{1, 2, 3, 4, 5, 6, 7, 8, 9, 10\}$.

Figure 10 (a) provides a visual representation of the impact of accuracy and the hyperparameter $k$ on the planning time. For each pair of accuracy and $k$, we run 10 experiments with different random seeds. The mean time and standard deviation of the average step are plotted in the figure, providing a comprehensive view of how these factors influence the planning time. We also present the time required for the agent planning when using $\mathcal{A}$ only and $\mathcal{T}$ only as in normal agent planning to show the lower bound and upper bound of speculative planning.

It is evident that higher accuracy in $\mathcal{A}$ results in shorter planning time. Very low $k$ (such as $k = 1, 2, 3$) leads to slower agent planning, regardless of $\mathcal{A}$'s accuracy. For other $k$ values, as the accuracy increases, the impact of $k$ becomes more clear: higher $k$ leads to shorter agent planning time. However, when the accuracy is low, the impact is less clear.

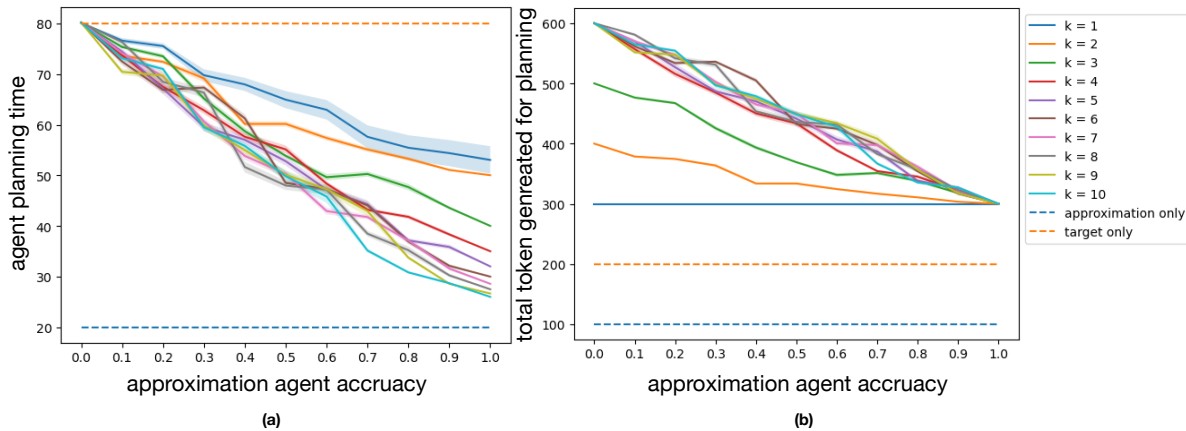

**Figure 10:** (a) Relationship between agent planning time, $\mathcal{A}$'s accuracy, and $k$ (b) Relationship between total token generated, $\mathcal{A}$'s accuracy, and $k$

Figure 10 (b) showcases the impact of accuracy and the hyperparameter $k$ on the total number of tokens generated during the planning process. In addition, the figure presents the tokens generated when using only $\mathcal{A}$ and when using only $\mathcal{T}$. There are two obvious trend: (1) higher accuracy in $\mathcal{A}$ generally results in a smaller number of tokens generated regardless of $k$ except in the trivial case when $k = 1$ (2) lower $k$ leads to a smaller number of tokens to be generated, especially when $k$ is small in the value range of $\{1, 2, 3, 4\}$; otherwise is impact is less clear.

In the second series of experiments, we study the impact of $\mathcal{A}$'s speed and accuracy on planning time and generated tokens: We experiment on speed in different values: $\{1, 2, 3, 4, 5, 6, 7, 8\}$ and accuracy in values $\{0.0, 0.1, 0.2, 0.3, 0.4, 0.5, 0.6, 0.7, 0.8, 0.9, 1.0\}$. Here we set $k$ to be 5. Figure

11 (a) demonstrates the change of planning time: (1) smaller speed values lead to smaller planning time, *i.e.*, quicker planning, regardless of the accuracy of $\mathcal{A}$, and (2) better accuracy also leads to quicker planning, regardless of speed.

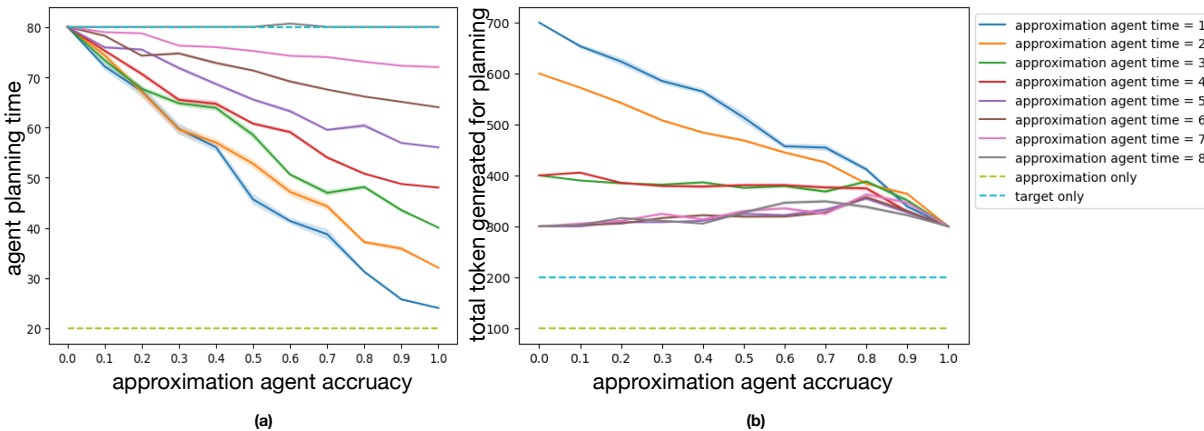

**Figure 11:** (a) Relationship between time and $\mathcal{A}$'s accuracy and speed (b) Relationship between time and $\mathcal{A}$'s accuracy and speed

Figure 11 (b) demonstrates the effect on total token generated. When the speed is very quick, higher accuracy monotonically reduces the total token generated. When the speed is around half of the speed of $\mathcal{T}$, accuracy does not have much impact on total tokens until it gets very high. When the speed is very slow (more than half of that of the target process), the total token generated first increases and then decreases as accuracy improves.

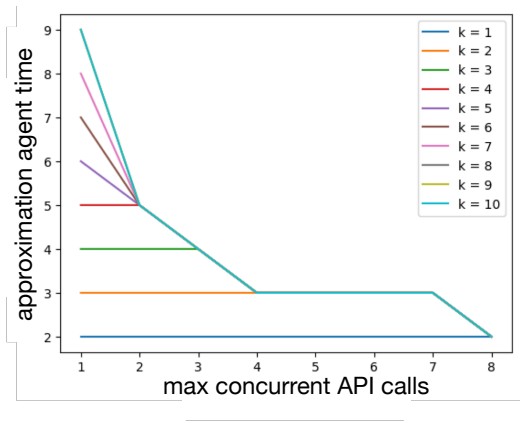

**Figure 12:** Relationship between maximum concurrent rate required and $\mathcal{A}$'s speed and $k$

In the third series of experiments, we investigate the impact of user interruptions on time efficiency. We conduct simulations with varying interruption times. We assume that the user is actively monitoring the agent planning process and has a patience level between the speeds of the approximation agent $\mathcal{A}$ and the target agent $\mathcal{T}$, which assumption is made based on (1) $\mathcal{A}$ is designed to be an efficient agent (2) if the user's patience exceeds the speed of $\mathcal{T}$, no interruptions would occur.

For this simulation experiment, we set $k = n = 10$ and the accuracy of $\mathcal{A}$ to be 0.5. The user is permitted to interrupt between 0 and 10 times. Each user interruption may occur randomly after waiting periods ranging from 1 to 5 seconds following the presentation of $\mathcal{A}$'s result. For each number of user interruptions, we conduct 5 simulations.

The results of this simulation are presented in Figure 13, which displays the mean stepwise generation time along with the standard deviation. As anticipated, an increase in user interruptions reduces the overall latency of the system.

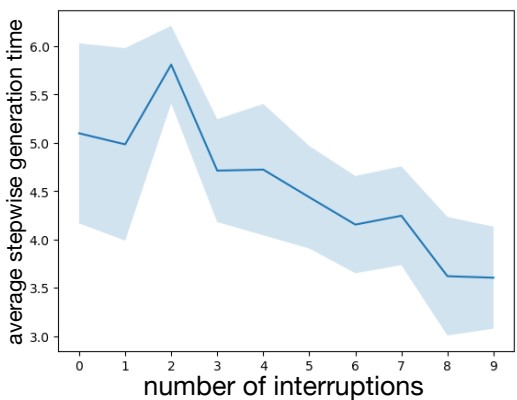

**Figure 13:** Relationship between the number of user interruption simulation and stepwise generation time

## D IMPLEMENTATION DETAILS ON TRAVELPLANNER

To evaluate the final plans generated by normal agent planning and speculative planning, we adopt the metrics Delivery Rate and Commonsense Constraint Micro Pass Rate (the only two metrics with non-trivial results):

**Delivery Rate**   assesses whether agents can successfully deliver a final plan within a limited number of steps. Falling into dead loops, experiencing numerous failed attempts, or reaching the maximum number of steps (30 steps in our experimental setting) will result in failure.

**Commonsense Constraint Micro Pass Rate**   Commonsense Constraint Pass Rate comprises eight commonsense dimensions, which evaluates whether a language agent can incorporate commonsense into their plan without explicit instructions. The Macro Pass Rate indicates the ratio of passed constraints to the total number of constraints.

Below are the three set of results:

**Setting 1**   normal agent planning:
Delivery Rate: 55.6% Commonsense Constraint Micro Pass Rate: 48.6%

speculative planning:
Delivery Rate: 55.6% Commonsense Constraint Micro Pass Rate: 41.7%

**Setting 2**   normal agent planning:
Delivery Rate: 55.6% Commonsense Constraint Micro Pass Rate: 41.7%

speculative planning:
Delivery Rate: 55.6% Commonsense Constraint Micro Pass Rate: 34.7%

**Setting 3**   normal agent planning:
Delivery Rate: 55.6% Commonsense Constraint Micro Pass Rate: 54.3%

speculative planning:
Delivery Rate: 55.6% Commonsense Constraint Micro Pass Rate: 48.6%

## E ANALYSIS OF TIME EFFICIENCY IMPROVEMENT BREAKDOWN FOR TRAVELPLANNER

Figure 14, 15, and 16 present the results for the TravelPlanner dataset. In TravelPlanner, the distribution of datapoints based on accuracy is flatter (we only show accuracy levels where the proportion of datapoints exceeds 2% to avoid excessive randomness). In all three settings, the time efficiency improvement can achieve more than 40% when the accuracy exceeds approximately 0.4.

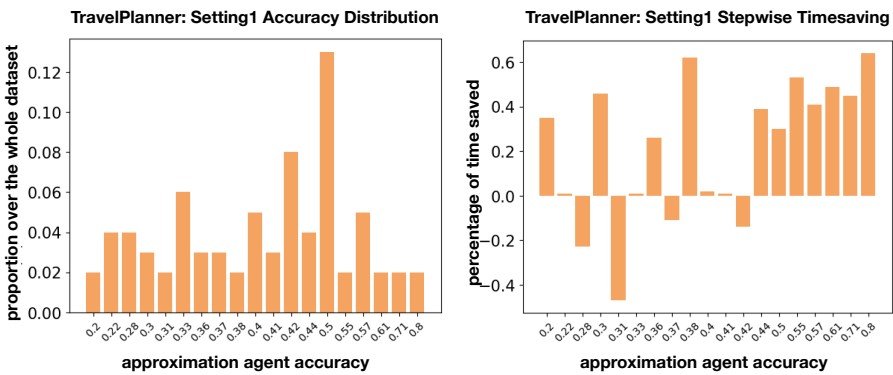

**Figure 14:** Distribution of $\mathcal{A}$'s accuracy in Setting 1 and corresponding time efficiency improvement

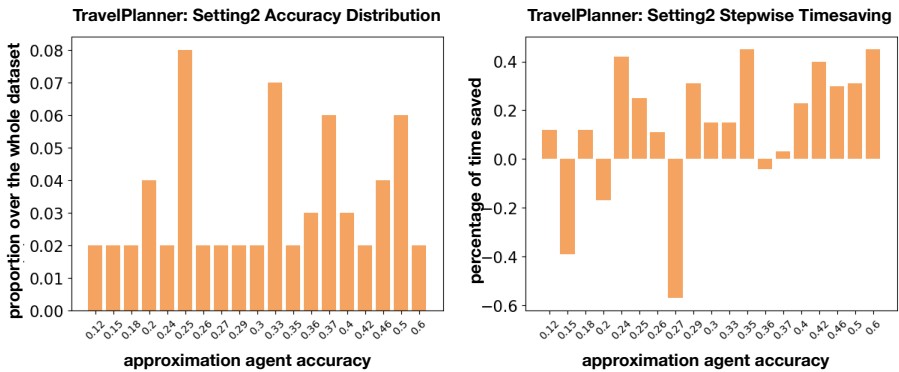

**Figure 15:** Distribution of $\mathcal{A}$'s accuracy in Setting 2 and corresponding time efficiency improvement

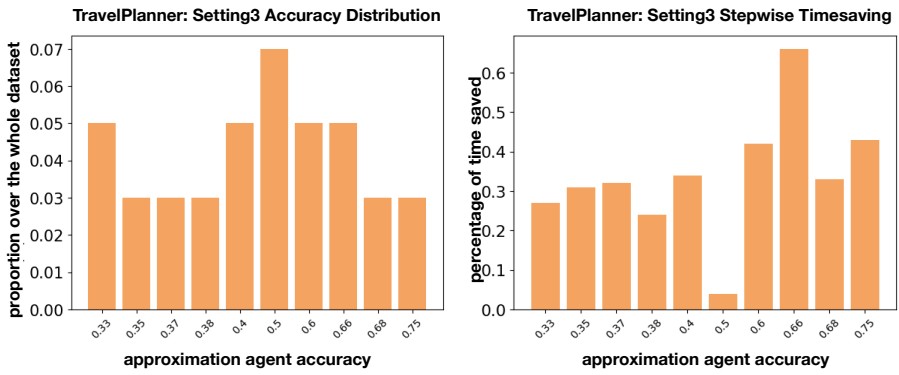

**Figure 16:** Distribution of $\mathcal{A}$'s accuracy in Setting 3 and corresponding time efficiency improvement

## F    ANALYSIS OF USER INTERACTION

One of the motivations behind Interactive Speculative Planning is users' patience. Numerous studies (Horvitz, 1999; Barron et al., 2004; Simpson et al., 2007; Carr et al., 1992) have demonstrated the physiological and psychological impacts of interaction delays on human-computer interaction. Therefore, we aim to quantitatively study how speculative planning enhances user experience by analyzing the frequency with which a user may become impatient and desire to interact with or interrupt the system.

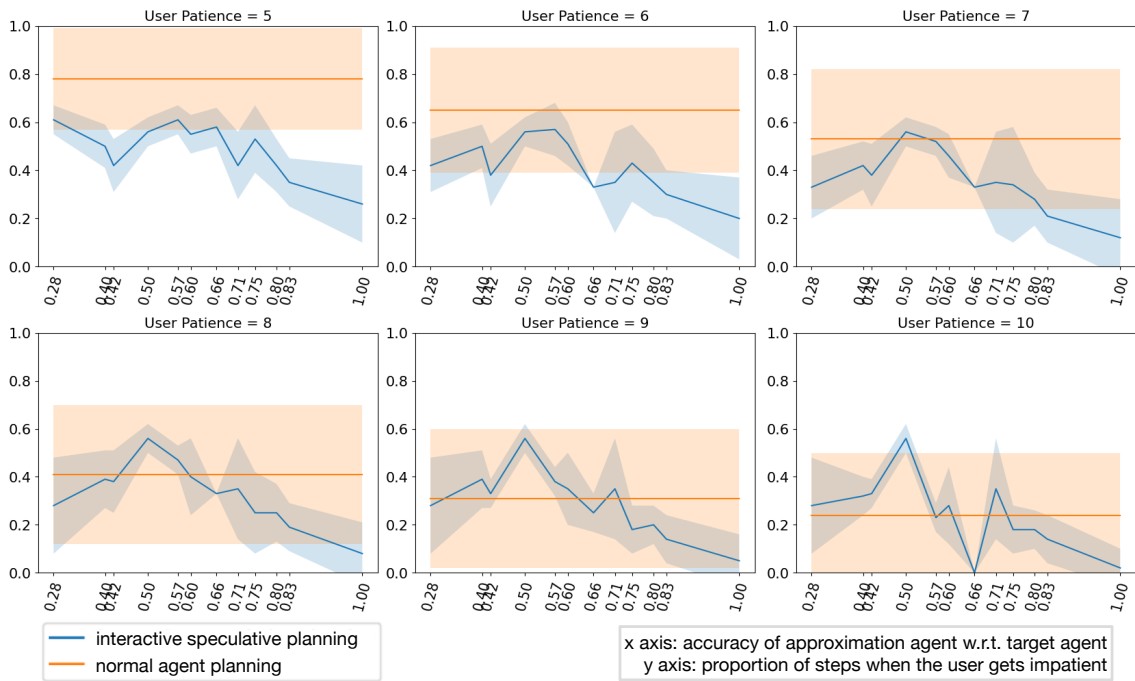

**Figure 17:** Number of Potential User Interruptions with Setting 1 on OpenAGI dataset and corresponding normal agent planning

For the quantitative study, we use the OpenAGI dataset with Setting 1, 2, and 3 as examples[2]. We collect statistics, including the mean and variance, on the number of user interruptions that may occur due to impatience by simulating users with different impatience thresholds. For Settings 1 and 2, we simulate users with impatience thresholds of 5, 6, 7, 8, 9, and 10 seconds. For Setting 3, which takes a much longer time to run, we simulate users with impatience thresholds of 11, 13, 17, 19, and 21 seconds. We also collect statistics for normal agent planning for comparison. For each setting, we provide a series of six figures. In each figure, the x-axis represents $\mathcal{A}$'s accuracy, and the y-axis represents the proportion of steps for which the user may become impatient. Each figure demonstrates the number of times the user may become impatient and interact with the system under Interactive Speculative Planning and normal agent planning, with respect to groups of data points with different levels of $\mathcal{A}$'s accuracy.

Figures 17, 18, and 19 represent the results for Settings 1, 2, and 3, respectively. As expected, Interactive Speculative Planning exhibits more observable differences for more impatient users.

## G    LIMITATIONS AND FUTURE DIRECTIONS

Interactive Speculative Planning represents the first attempt at co-designing an efficient agent system alongside an active user interface. Consequently, it is imperfect in many aspects, and there are numerous future directions to be explored:

---

[2]We do not adopt Setting 4 here as the average stepwise time for both normal agent planning and speculative planning is too short.

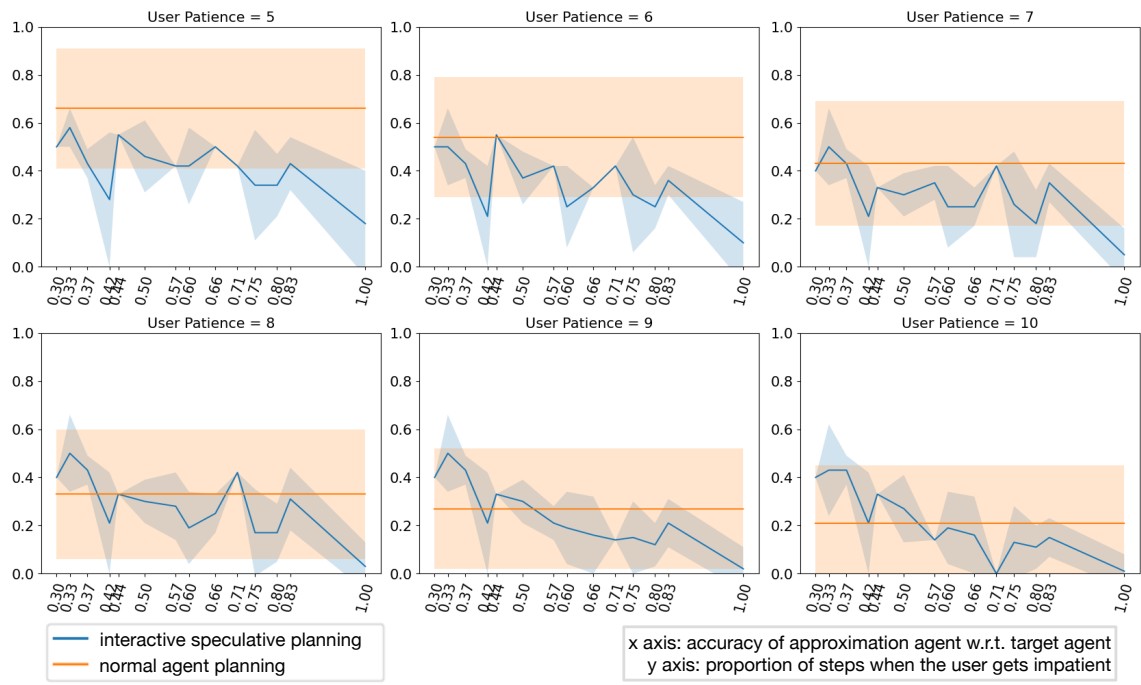

**Figure 18:** Number of Potential User Interruptions with Setting 2 on OpenAGI dataset and corresponding normal agent planning

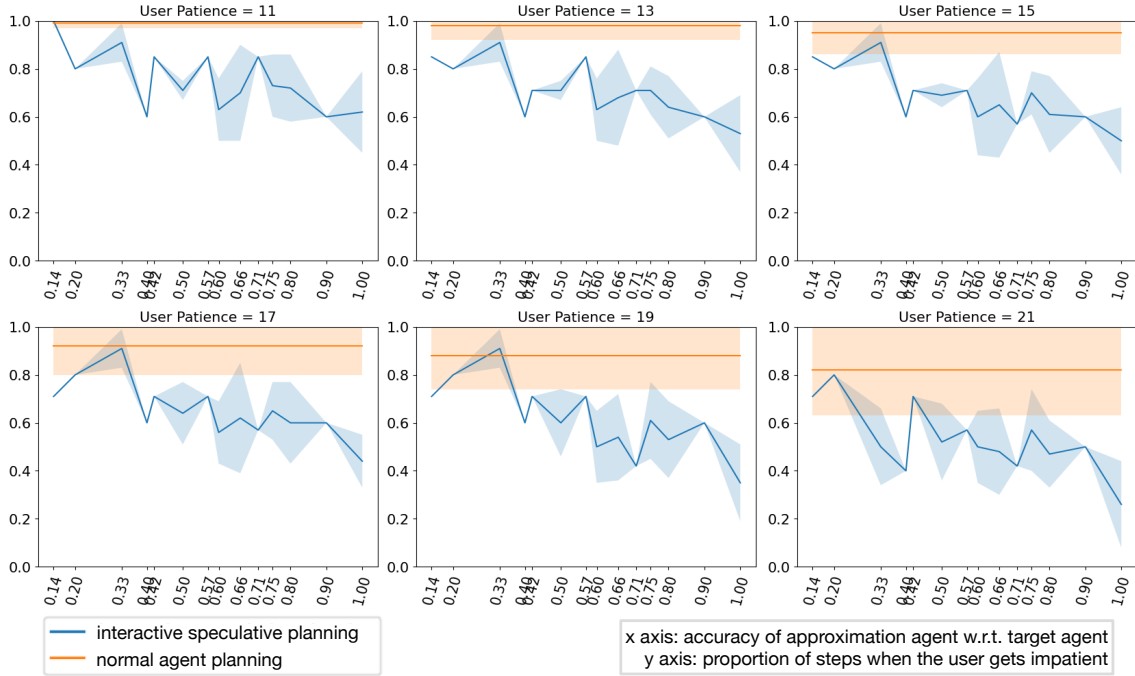

**Figure 19:** Number of Potential User Interruptions with Setting 3 on OpenAGI dataset and corresponding normal agent planning

**Spectre** Spectre (Mcilroy et al., 2019; Kocher et al., 2020) refers to vulnerabilities and attacks involved in speculative execution (Kocher et al., 2020; Gabbay & Mendelson, 1996; Nightingale et al., 2005), a hardware feature that improves processor performance by predicting a program's

future execution and executing instructions ahead of the current instruction pointer. This concept is analogous to speculative planning. However, speculative execution is known to create security vulnerabilities that allow attackers to access sensitive data. In our algorithm, the execution of $\mathcal{A}$'s steps that are unverified or inconsistent with $\mathcal{T}$'s steps also introduces vulnerabilities. Studies (Hua et al., 2024) have shown that smaller and weaker models are more prone to unsafe and untrustworthy actions. Therefore, the applicability of Interactive Speculative Planning in its current form should be constrained to non-high-stakes areas.

To address the security vulnerabilities associated with speculative execution in Interactive Speculative Planning, several solutions can be implemented:

1. Human-in-the-Loop Verification: Incorporate human-in-the-loop mechanisms to double-check the security of $\mathcal{A}$'s actions before execution. This approach leverages human oversight to ensure that potentially unsafe actions are identified and mitigated before they can cause harm.

2. Isolated Execution Environments: Execute $\mathcal{A}$'s actions in isolated environments, such as Docker containers. This isolation ensures that any potentially malicious or untrustworthy actions are contained and do not affect the broader system or access sensitive data.

By implementing these solutions, we can enhance the security of Interactive Speculative Planning, making it more robust and suitable for a wider range of applications, including those in high-stakes environments.

**Effectiveness of step-by-step comparison**    In the current version of speculative planning, we employ a simple and straightforward method to determine whether an action proposed by $\mathcal{A}$ can be accepted: exact match. However, it is widely recognized that completing a task often involves multiple different paths and plans, and "difference" does not necessarily imply "incorrect." There are two types of differences to consider: (1) different surface strings may refer to the same step, and (2) different steps may refer to two acceptable paths for the planning. Therefore, a step-by-step exact match judgment for whether $\mathcal{A}$'s output is accepted is overly aggressive and inefficient, as it essentially decreases the accuracy of $\mathcal{A}$.

Consequently, there is a need for more sophisticated methods to relax the conditions for accepting actions from $\mathcal{A}$. Potential solutions may include:

1. Step-by-Step Relaxed Exact Match: While still performing step-by-step checks, this approach does not enforce an exact match between $\mathcal{A}$'s result and $\mathcal{T}$'s result. Instead, it allows for some degree of flexibility in what constitutes a match.

2. Postponed Judgment: Instead of performing step-by-step checks, $\mathcal{T}$ will judge whether a sequence of $n$ steps proposed by $\mathcal{A}$ is within an acceptable range. This approach allows for a more holistic evaluation of $\mathcal{A}$'s proposals.

By implementing these solutions, we can enhance the effectiveness and efficiency of speculative planning, making it more adaptable to the variability inherent in task completion.

**Balancing time and cost efficiency in speculative planning**    The current version of speculative planning may incur a high additional cost. Therefore, balancing time efficiency and cost efficiency becomes a critical topic. Several methods can be employed to reduce the cost effectively:

1. Utilize a Cost-Effective $\mathcal{A}$: Employ a cheaper yet well-functioning $\mathcal{A}$. This agent can be trained through knowledge distillation from $\mathcal{T}$, thereby improving performance while maintaining a smaller size.

2. Enhance the Approximation-Target Judgment Method: Implement a more sophisticated approximation-target judgment method to improve the perceived accuracy of $\mathcal{A}$. This approach ensures that $\mathcal{A}$'s outputs are more reliably accepted, reducing the need for costly re-evaluations by $\mathcal{T}$.

By implementing these methods, we can achieve a better balance between time efficiency and cost efficiency in speculative planning.

**Limitations of the Current User Interface**    As mentioned in the algorithm design section, although the current user interface can handle active user input, these interactions and interruptions must be made "on time." Specifically, users cannot change the result once it is fully presented in the user interface. This limitation means that users must closely monitor the algorithm's progress, and if they miss the opportunity to intervene, there is no way to go back and make modifications. In the future, the user interface should support backtracing to allow users to revisit and modify previous steps, enhancing the flexibility and usability of the system.

