# OpenReview forum: "Interactive Speculative Planning: Enhance Agent Efficiency through Co-design of System and User Interface"
_ICLR.cc/2025/Conference — ICLR 2025 Poster_

### Official Review · Reviewer_SCq5 · 2024-10-28

**Soundness:** 3
**Presentation:** 2
**Contribution:** 2
**Rating:** 6
**Confidence:** 4

**Summary:**

The paper introduces a speculative planning framework to improve the time efficiency of agent-based task planning. The paper tackles reducing latency by implementing two agent types. An approximation agent generates initial, potentially incorrect steps, which the more accurate target agent validates. This setup allows speculative planning to proceed concurrently, ideally saving time without sacrificing accuracy. Through experiments with two agent planning benchmarks, OpenAGI and TravelPlanner validation and metrics such as time efficiency, API costs, etc, the paper demonstrates the efficiency gains. It also allows users to get involved in the planning process. However, this aspect is speculative and has not been tested.

**Strengths:**

1) In terms of writing, the paper is well-organised and engaging. The planning algorithm is discussed well -- the speculative planning process is generally explained well. However, a more detailed discussion with a specific k-value around Fig 1 would be helpful in understanding the various scenarios and how these result due to the accuracy of A. For example, it will help the reader speculate how A's accuracy may impact the number of times replanning may be needed, etc.

2) The paper evaluates various settings (e.g., ReAct, CoT, MAD, and DG), revealing a good level of depth in its potential impacts. The experiments are thorough, with 4 settings and 2 planning benchmarks (OpenAGI and TravelPlanner). Metrics such as time efficiency, API costs, and stepwise validation provide a good depth of analysis, both positive and negative aspects.

3) In terms of novelty, while speculative planning may not be universally applicable, its potential to accelerate high-latency planning tasks is valuable for the AI community, especially for applications requiring real-time or near-real-time responses.

**Weaknesses:**

1) One downside, not fully addressed or discussed in the paper, is the increased cost. The paper largely supports its claims of time efficiency gains, presenting results indicating speculative planning can save time, especially in complex planning settings. However, cost savings are not fully supported, as all speculative settings incur higher costs. While this increase is expected, as A and T operate concurrently, the paper only highlights positives and does not discuss costs. To balance the discussion, the authors must discuss both sides: pros and cons in Sections 4.1 and 4.2. While time savings are the focus, I assume real-world deployments must balance cost and efficiency, particularly if A has very low accuracy and frequently diverges from the steps produced by T, which increases wasted resources. Therefore, an open discussion about this limitation must be expressed in the same units as efficiency is justified (%) and possibly what strategies may allow users to reduce the cost.

2) Although the title suggests a "co-design" with user interface considerations, the paper provides limited insights into UI and user interactions. I find this aspect genuinely troublesome. The paper is technical. Even the UI elements, such as producing the steps in the correct order etc., are technical contributions. This is not co-design from an HCI perspective. I also have a few more questions on this aspect, which are provided in the Questions section later.

3) The framework relies on hyperparameter k. In the experiments, the authors used k=4 -- why? The appendix does provide some insight into the sensitivity of this parameter. Still, since k is central to the scheme, providing some guidance on selecting k would be beneficial. For example, could we employ dynamic tuning of k, etc?


Minor issues:
- 048: and The sequential
- In Figure 1, please briefly state the reason for using Venmo as an example.
- Line 100: Statements like "This strategy [[potentially]] reduces the time a target agent" suggests either that this gain is not universal or that the authors have doubts about their results. If there are caveats regarding when we expect reduced time, then that needs to be made explicit.
- I did not find the Venmo case study or the diagrams helpful.

**Questions:**

Since the cost is one of the main issues:

1) Under what circumstances should each evaluation metric (e.g., accuracy vs. speed vs. cost) be prioritised, and how might these impact the framework's configuration, e.g., in terms of k? The cost increase is above 60% in some cases. For example, do the authors believe there is a case for solely focusing on cost and not efficiency and vice versa?

2) Do the authors have insight into reducing costs to make the approach more appealing? The appendix states: "...Implement a more sophisticated approximation-target judgment method..." --- What does this look like?

Regarding the ability of the users to intervene, etc.:

3) How do you envision training end-users on the speculative planning interface, particularly in complex tasks where they may need to understand and manage concurrent agent processes?

4) Also, given that user intervention is integral to your framework, how would you design the interface to facilitate timely and accurate user decisions?

Regarding resource-constrained environments:

5) What considerations should be explored for scaling the speculative planning framework in resource-constrained environments, especially where concurrent instances of T may lead to bottlenecks?
6) Since the approach's success largely depends on the accuracy of A, what do the results and analysis of k (in the appendix) indicate in terms of any minimum thresholds for A's accuracy?

---

> ### Author Response · Authors · 2024-11-19
> **Rebuttal for Weaknesses 1 & 2**
>
> We would like to extend our sincere gratitude for the invaluable time and effort you've dedicated to reviewing our manuscript and for providing us with detailed feedback.
>
> Below are our replies to the concerns you raised:
>
> ---
>
> > ***W1: One downside, not fully addressed or discussed in the paper, is the increased cost.***
>
> Thank you for pointing out this important aspect. We appreciate your feedback and agree that a balanced discussion of both the advantages and limitations of our approach is essential.
>
> **Firstly, we will definitely add more information and analysis on the increased cost in Sections 4.1 and 4.2**. In our main results table in section 4.1 and 4.2, we present the time saved but also the increased cost in terms of total tokens generated, the number of concurrent API calls, and the overall change in cost. We will add an analysis of the increasing cost. Here is an example analysis that we can/will present in the updated version of the paper:
> ```
> In the first and second settings, the computation time decreased by approximately 20% and 30%, respectively, at the cost of a ~70% increase in monetary cost. In the third setting, the computation time decreased by about 40%, at the cost of a 37% increase in monetary cost. This is because, in the third setting, multi-agent discussion is itself a very expensive prompting method. In the last setting, where we use GPT-3.5 as the approximation agent, the computation time decreased by about 20% at almost no additional cost due to the very cheap nature of GPT-3.5. Thus, we can see that the increase in cost can be mitigated if the prompting method is simpler or if the approximation agent’s backbone model is cheaper.
> ```
> **Secondly, we would like to direct your attention to Appendix C2, where we have extensively analyzed the increased cost.** In this section, we cover:
>
> (1) The best and worst cases of total tokens generated in the system, which directly correspond to the cost.
>
> (2) The best and worst cases of concurrent API calls required in the system.
>
> We will move some of this detailed analysis back to the main body of the paper to ensure that the discussion is comprehensive and balanced. Thank you again for your valuable feedback.
>
>
> > ***W2: Although the title suggests a "co-design" with user interface considerations, the paper provides limited insights into UI and user interactions. The paper is technical.***
>
> Thank you very much for raising this important point. We want to elaborate the standpoint of the paper a bit more.
>
> **Our paper is one of the first to incorporate active user engagement into agentic framework design**, and we consider latency to be a key aspect that users care about from an HCI perspective. As an initial effort in this direction, our focus has been on establishing the technical foundation to enable and integrate user engagement into the system.
>
> By building a system that accelerates agentic task planning and can handle active user interactions, **we are laying the groundwork for broader design considerations**. This includes potential user studies on **how much users want to accelerate the process** (which we can control by setting the approximation agent and the parameter k), **how much information users want to see** (the whole generation process, the final result, or something in between), and **other possible implementation of other technical user-interaction mechanisms** such as a roll-back mechanism. This technical groundwork is essential for paving the way for more user-centric agent framework designs in the future. We will work on enhancing our discussion to better highlight the HCI elements and provide more insights into how the technical contributions support and inform the UI and user interaction design.
>
> Thank you again for your valuable feedback.

---

> ### Author Response · Authors · 2024-11-19
> **Rebuttal for Weakness 3**
>
> > ***W3: In the experiments, the authors used k=4 -- why? Since k is central to the scheme, providing some guidance on selecting k would be beneficial. For example, could we employ dynamic tuning of k, etc?***
>
> Thank you for raising this important point. **The choice of k=4 in our experiments is a heuristic decision aimed at balancing cost and acceleration speed.**
>
> Based on the accuracy breakdown analysis presented in the appendix, we observe that in the OpenAGI dataset, settings 1, 2, and 3 have an approximation agent with around 0.7 accuracy, while setting 4 has around 0.5 accuracy. **We selected k such that the probability of k sequential steps being correct is not too low.** For an accuracy of 0.7, the probability of three adjacent steps being correct is 34%, the probability of four adjacent steps being correct is 24%, and the probability of five adjacent steps being correct is 17%. Therefore, we subjectively chose k=4 to strike a balance.
>
> **Regarding the idea of dynamic k**, this is indeed a very interesting and promising approach. Some papers on speculative decoding, such as [1] have explored the possibility of a dynamic drafting step (k) based on the observation that the difficulty of predicting the next token varies across different contextual scenarios. They use a confidence threshold to stop the drafting model from further generation once the confidence score drops below it. A similar idea could be applied to speculative planning, where k could be dynamically adjusted based on the context and confidence levels.
>
> We appreciate your feedback and will consider incorporating dynamic tuning of k in our future work to enhance the flexibility and efficiency of the framework in the future.
>
> [1] Kangaroo: Lossless Self-Speculative Decoding via Double Early Exiting
>
> **More detailed analysis on how to choose k can be found in https://openreview.net/forum?id=BwR8t91yqh&noteId=5v5YExOc78**

---

> ### Author Response · Authors · 2024-11-19
> **Reply for Questions 1 - 4**
>
> > ***Q1: In Figure 1, please briefly state the reason for using Venmo as an example.***
>
> **Venmo was chosen as an example simply because it is in widespread use**, making it a relatable and accessible illustration for readers. However, if it is not very professional to use a specific App name in the paper, **we could certainly change the example to a "bank app" or another universally recognized application to maintain a more professional and generic tone**. We can make this adjustment in the final version of the paper.
>
> > ***Q2: Line 100: Statements like "This strategy [[potentially]] reduces the time a target agent" suggests either that this gain is not universal or that the authors have doubts about their results.***
>
> Sorry for the confusion. We use the word “potentially” because **in the worst case scenario** where the approximation makes a mistake for every single step, **the running time of the system will be exactly the time of normal agent planning**. Therefore, though very unlikely, it is possible that there is no time reduction at all. This is why we use the word potentially, just to cover this very low possibility.
>
> > ***Q3 (a): Under what circumstances should each evaluation metric (e.g., accuracy vs. speed vs. cost) be prioritised, and how might these impact the framework's configuration, e.g., in terms of k?***
>
> **For accuracy**: the prioritization of accuracy does not affect k, as this method guarantees the performance to be the same as normal agent planning. Thus no matter what k is, accuracy will not be sacrificed (unless some lossy matching method is used)
>
> **For speed**: as presented in **Appendix C4. Figure 10 (a)**, higher k always leads to quicker system. And thus if speed is prioritized, we should use very large k.
>
> **For cost**: as presented in **Appendix C4. Figure 10 (b)**, a lower k always leads to a lower cost. And thus if cost is prioritized, we should use very small k.
>
> More detailed analysis on how to choose k can be found in https://openreview.net/forum?id=BwR8t91yqh&noteId=5v5YExOc78
>
> > ***Q3 (b): do the authors believe there is a case for solely focusing on cost and not efficiency and vice versa?***
>
> Yes, we believe there are cases where it is appropriate to focus solely on either cost or efficiency, depending on the specific context and user needs.
>
> **Prioritizing Efficiency**: In scenarios such as customer service chatbots, quick response times are crucial for user satisfaction and engagement. Users expect immediate assistance, and delays can lead to frustration or loss of trust. In such cases, it makes sense to prioritize efficiency even if it results in higher costs.
>
> **Prioritizing Cost**: Conversely, there are many situations where time efficiency is less critical, and minimizing cost becomes the primary concern. For example, when designing a travel plan for a user in a non-urgent context, the user can attend to other tasks while waiting for the plan to be generated.
>
> > ***Q4: Do the authors have insight into reducing costs to make the approach more appealing?***
>
> Thank you for raising this important question. There are several notable approaches in the literature that address cost reduction in agent planning:
>
> **EcoAssistant**: The paper titled "EcoAssistant: Using LLM Assistant More Affordably and Accurately" employs a model cascade to reduce planning costs by initially using a smaller, more efficient model. While this approach saves cost, it does so at the expense of increased latency.
>
> **System-1.x**: The paper "System-1.x: Learning to Balance Fast and Slow Planning with Language Models" fine-tunes a controller, a System-1 Planner, and a System-2 Planner based on a single LLM. This system uses the controller to decide whether to use System-1 or System-2 for planning specific steps, thereby reducing costs. However, this method requires model training on specific tasks.
>
> **Online Speculative Decoding**: The paper is about speculative decoding but I think the idea can be adopted to speculative planning. Online Speculative Decoding proposes tuning the approximation LLM based on feedback from the larger target LLM. This approach enhances the accuracy of the approximation model and reduces latency through online learning. A similar idea could be applied in our context to improve accuracy of the approximation agent and thus reducing the cost (fewer wasted processes) as well as latency.
>
> Currently, I do not have any other brand-new ideas for reducing both latency and cost beyond what these models have discussed. And in general, I think model tuning/training may be unavoidable to reduce both latency and cost. However, this is at the top of my to-do/to-think list, and I am actively exploring potential solutions.
>
> Thank you for your insightful feedback. We will continue to investigate ways to optimize both cost and latency in our future work.

---

> ### Author Response · Authors · 2024-11-19
> **Reply for Question 5 - 7**
>
> > ***Q5: How do you envision training end-users on the speculative planning interface, particularly in complex tasks where they may need to understand and manage concurrent agent processes?***
>
> We believe that **end-users won't require specialized training** to use the speculative planning interface. In our UI design, the underlying concurrent agent processes are abstracted away, presenting the planning as a standard multi-agent system involving two agents. From the user's perspective, it's a sequential interaction where an approximation agent suggests planning steps, and a target agent agrees or disagrees. Much like speculative decoding, users remain unaware of the backend mechanics. Our speculative planning approach is seamlessly integrated into the UI (**see page 6**), ensuring users can focus on their objectives without needing to understand or manage the concurrent processes behind the scenes.
>
> > ***Q6: Also, given that user intervention is integral to your framework, how would you design the interface to facilitate timely and accurate user decisions?***
>
> **In page 6, we discuss when and which agent’s result to present to the users on the user interface.** Specifically, the user interface only displays the approximation agent’s response based on a confirmed action trajectory. For instance, if the approximation agent quickly generates two sequential actions before the target agent confirms the first one, only the first action generated by the approximation agent will be shown. The system will then wait for the target agent's result on the first step before presenting the next message, which is the target agent’s result on the first step. If the target agent’s result aligns with the approximation agent’s result, the second step computed by the approximation agent will be presented, as it is based on a correct action trajectory.
>
> **Consequently, the user will see pairs of (approximation agent’s generated response at step i, target agent’s generated response at step i) sequentially. This design aims to reduce user confusion and enhance transparency, ensuring a clear and coherent interaction experience.**
>
> Another question worth asking is what content generated by each agent we should provide to the user: **should we display only the final result for each step or the entire thinking process?** In the former case, there would be a longer wait for the step to be presented; in the latter, the volume of text generated could be overwhelming. This decision also influences whether we are able to and whether we should expect immediate user interruptions. In the current system design, we choose to present only the final decided step, which reduces the cognitive load on the user. But further exploration on how much information we should provide to the users and how to design the corresponding user interface can be and should be discussed.
>
> > ***Q7: What considerations should be explored for scaling the speculative planning framework in resource-constrained environments, especially where concurrent instances of T may lead to bottlenecks?***
>
> Thank you for raising this important question. **If I understand correctly, you're asking how we can scale the speculative planning framework in resource-constrained environments when multiple users are requesting the agent simultaneously or in very close time**.
> To address this challenge, several considerations should be explored:
>
> (1) **Overall Latency**: We need to ensure that the total waiting time for all users is minimized. Efficiently distributing resources can help reduce latency and improve the overall user experience.
>
> (2) **Fairness**: It's important to prevent scenarios where some users experience excessively long wait times due to optimizations aimed at minimizing total latency. We should strive for a balanced approach that avoids disproportionately disadvantaging any user request.
>
> (3) **Cost Management**: Balancing the latency and cost, though in situations with limited concurrent instances of T, the cost is also upperbounded as we cannot have large k for most requests.
>
> (4) **Request Prioritization**: Recognizing that some tasks are more urgent and require lower latency, we should implement a priority system. Urgent requests can be given higher priority, while less time-sensitive tasks can be scheduled accordingly.
>
> By carefully considering these factors, we can develop strategies to effectively scale the speculative planning framework, ensuring it remains efficient and fair even in resource-constrained settings.

---

> ### Author Response · Authors · 2024-11-19
> **Reply for last question**
>
> > ***Q: Since the approach's success largely depends on the accuracy of A, what do the results and analysis of k (in the appendix) indicate in terms of any minimum thresholds for A's accuracy?***
>
> Thank you for your insightful question regarding the minimum thresholds for A's (the approximation agent's) accuracy in relation to k. Appendix C4 provides a comprehensive overview of how the speed of A, the accuracy of A, and the value of k interact.
> But here, To offer a more concrete method for selecting hyperparameters, we introduce a user-preference parameter $\alpha$, where $0\leq\alpha\leq\inf$. This This parameter represents the trade-off between time saved and cost increased, both measured in percentages. We define $\alpha$ as
> $$\alpha = \frac{\text{time of NAP}/\text{time of SP}}{\text{cost of NAP}/(\text{cost of target agent using NAP} + \text{cost of approximation agent using NAP})}$$
> where NAP is the abbreviation of normal agent planning and SP is the abbreviation of speculative planning. We choose the cost baseline to be cost of approximation agent ($A$) using NAP + cost of target agent ($T$) using NAP, as this represents the minimum possible number of tokens the system will generate. If a user cannot afford this token generation cost, they should not use such a system.
>
> A higher $\alpha$ indicates that we require time increase percentage to be >= the cost increase percentage; a lower $\alpha$ means we can bear very high cost or we can bear very large latency.
>
> To determine the configuration of the speculative planning system, we consider two primary questions:
>
> 1. **Given specific configurations of $T$ and $A$, and a certain preference value $\alpha$, how do we determine the possible values of k?**
>
> 2. **Given a specific $T$ and a fixed k (possibly determined by available resources), and a certain preference value $\alpha$, how do we determine the configurations of $A$?**
>
> To answer the two questions, we adopt the simulation setting we use in Appendix C4:
>
> (1) $T$ (Target Agent): Takes 8 seconds per step, generates 30 tokens per step.
> (2) Plan: Consists of 10 steps.
>
> ---
>
> **For question 1:** Let's explore for different configurations of $A$ (accuracy, time per step), what k is required given a specific threshold of $\alpha$.
>
> 1. $A$ with (0.5 accuracy, 2 seconds per step):
>
> If $\alpha  = 1.2$, possible k values to obtain an $\alpha$ value higher than 1.2 are 2, 3, 8. This suggests we can either proceed slowly with limited k to minimize token waste or opt for a higher k to save time while controlling waste. A multi-objective trade-off is required here.
>
> 2. $A$ with (0.6 accuracy, 2 seconds per step):
>
>  If $\alpha  = 1.2$, possible k values can range from 2 to 10
>
> 3. $A$ with (0.7 accuracy, 2 seconds per step):
>
> If $\alpha  = 1.2$, possible k values can range from 2 to 10
>
> If $\alpha  = 1.5$, possible k values are 4, 5, 8, 9, 10
>
> 4. $A$ with (0.9 accuracy, 2 seconds per step):
>
> If $\alpha  = 1.2$, possible k values can range from 2 to 10
>
> If $\alpha  = 1.5$, possible k values can range from 2 to 10
>
> If $\alpha  = 2$, possible k values can range from 5 to 10
>
> **Basically, higher accuracy in $A$ allows for greater flexibility in choosing k while meeting the user's preference between time and cost.**
>
> ---
>
> **For question 2**, Given a specific $T$ and a fixed k (possibly determined by available resources), and a certain preference value $\alpha$, how do we determine the configurations of $A$?
>
> For example, with k = 5:
>
> 1. If $\alpha  = 1.2$, then the choices for $A$ are relatively broad. Possible configurations are
>
> (0.5, 1), (0.6, 1), (0.6, 2), (0.6, 3)
>
> (0.7, 1), (0.7, 2), (0.7, 3), (0.7, 4), (0.7, 5)
>
> (0.8, 1), …,(1.0, 5)
>
> Minimum required accuracy: 0.5
>
> 2. If $\alpha  = 1.5$, then the choices for $A$ narrow down. Possible configurations are
>
> (0.7, 1), (0.7, 2), (0.7, 3)
>
> (0.8, 1), (0.8, 2), (0.8, 3), (0.8, 4), (0.8, 5)
>
> (0.9, 1)…, (1.0, 4)
>
> Minimum required accuracy: 0.7
>
> 3. If $\alpha  = 2$, Options are limited to high-accuracy and faster configurations. Possible configurations are
>
> (0.8, 1), (0.9, 1), (0.9, 2)
>
> (1.0, 1), (1.0, 2)
>
> Minimum required accuracy: 0.8
>
> **So in general, as $\alpha$ increases, the minimum required accuracy of $A$ also increases, and the acceptable time per step decreases.**
>
>
> In our simulation experiments, we have not yet exhaustively explored all possible combinations of $A$ and k, as there are approximately 1,000 potential combinations. For each combination, we conduct 10 experiments to account for randomness. We are actively working on completing these simulations to provide clear and detailed guidance for users. Our goal is to help users configure the system effectively based on their time constraints and resource availability.

---

> ### Author Response · Authors · 2024-11-23
> **Further discussion and potential score increase?**
>
> Dear Reviewer SCq5,
>
> We sincerely appreciate the time and effort you've invested in reviewing our paper. Your insightful comments and thoughtful feedback are invaluable to us.
>
> We hope that our responses have addressed your questions and clarified any ambiguities. Please do not hesitate to reach out if you have any further questions!
>
> **We would be truly grateful if you would consider our clarifications and kindly reevaluate your score.**
>
> Thank you once again for your consideration.
>
> Best regards,
> the authors

---

> > ### Comment · Reviewer_SCq5 · 2024-11-26
> >
> > I thank the authors for their responses. Thank you for agreeing to incorporate a discussion on the approach's costs and providing insights into how these can be addressed. However, I'm not convinced with your discussion on W2 and Q5. We may "believe" that our designed system/UI "will" work," but that remains a hypothesis unless demonstrated. As such, I leave my score as it is.

---

> > > ### Author Response · Authors · 2024-11-26
> > > **Thank you for your reply**
> > >
> > > Dear reviewer SCq5,
> > >
> > > Thank you very much for your valuable feedback!
> > >
> > > **We acknowledge the importance of a comprehensive user study to fully understand the design needs for effective user interactions in agentic systems, whereas our work aims to provide a technical framework that supports various design choices.** By integrating the user as a critical component in the algorithm and addressing the core rescheduling challenge, we hope to lower the technical barrier and encourage more ML and HCI researchers to explore this user-centric LLM agent system direction.
> > >
> > > **Our proposed UI-level algorithm aims to present an intuitive and streamlined planning process.** The planning steps remain ordered to the user along with clear verification signals at the frontend, and all concurrent, backtracking, and pruning processes are handled in the backend. Therefore this approach is specifically designed to reduce the potential cognitive burden and minimize the need for extensive special training on end users.
> > >
> > > **While our contribution is technical and some hypotheses remain to be tested via real-world user studies, we believe these efforts are still valuable for enabling researchers to conduct follow-up studies and focus on core HCI problems.**
> > >
> > > We wanted to thank the reviewer again for these critical questions. We will revise the introduction to clarify the contributions of our work and enhance the discussion to better highlight how this framework can be incorporated into different UI designs.
> > >
> > > **In addition, I hope you find my replies to other questions such as the choice of k helpful :) We would be truly grateful if you would consider our clarifications and kindly reevaluate your score.**
> > >
> > > Best,
> > > the authors

---

### Official Review · Reviewer_y5dF · 2024-11-04

**Soundness:** 3
**Presentation:** 2
**Contribution:** 2
**Rating:** 6
**Confidence:** 3

**Summary:**

The paper presents an interesting approach to reduce the potential latency of LLM-based systems and an approach where user and user interruptions are build into system design. The approach involves two different agents: an faster but more error-prone approximation agent and a slower but more accurate target agent. Depending on the accuracy of approximation agent, the speed of system can be significantly faster , and in the worst case, comparable to a non-speculative planning agent.

**Strengths:**

Overall, the architecture of using a faster (but potentially inaccurate) agent and a slower (but possibly more accurate) in order to reduce the latency of the system is an interesting idea and can potentially work well for certain tasks with careful UI considerations.

**Weaknesses:**

1. The method applies to synchronous Human-AI interaction where a human is in the loop and waiting for agent response in real time. An entirely different application of Agents is asynchronous interaction (where a human may not be in the loop in real-time, e.g. perform this action every day at this time ..). I would encourage the authors to highlight this in order to position their work better.

2. I vehemently agree with the statement "A fully automated “blackbox” agent system with prolonged response delays is suboptimal for user experience". However, I believe the paper uses the term "user-centric" a bit loosely, I am not quite sure how the current method is more "user-centric". There are many places of contention:

a) Given approximation agent and target agent can (often) disagree, it means initial responses may be overwritten or multiple intermediate responses shown to user. Without careful UI design, this can lead to user confusion and lack of transparency on what is going on.
b) User in the loop workflow where user can actively take part in the decision making is absolutely necessary. However, in the current system, the window for interruption is basically the time for target agent to execute. The user can potentially take time to read the response from approximation agent, process it, decide if it needs to be revised manually and then potentially frame the revision in text or voice, all of which can take some amount of time.  I am failing to see how this is "user-centric" when user is hurried to interrupt within a window of short time.
c) I did not fully understand how this mechanism can work when the action execution can be anything of consequence in the user interaction (e.g. sending an email or request for money to split bill on a mobile pay application). This in practice would mean, if the approximation agent is wrong, the system would perform tons of incorrect actions that can not be reversed. I would like a more detailed limitations of where this approach can work and where it should not be applied.

More generally, there are established ways to study human factors aspects of Human-AI interaction, employing carefully controlled user studies. The current "theoretical approach" to human factors  is an interesting first step but leave a lot of questions unanswered that could potentially make or break the interaction.

**Questions:**

1. Could a theoretical focus on system latency lead to newer issues in the human-AI interaction?
2. Is this approach suitable for all tasks? A more detailed discussion would be useful.

---

> ### Author Response · Authors · 2024-11-18
> **Rebuttal for Reviewer 2 for Weakness 1, 2(a) & 2(b)**
>
> We would like to extend our sincere gratitude for the invaluable time and effort you've dedicated to reviewing our manuscript and for providing us with detailed feedback.
>
> Below are our replies to the concerns you raised:
>
> ---
>
> > ***W1: The method applies to synchronous Human-AI interaction where a human is in the loop and waiting for agent response in real time.***
>
> Thank you for raising this important point. Indeed, our method is designed for synchronous human-AI interaction, where a human is actively involved and waiting for real-time responses from the agent **because latency matters the most in real time scenarios.** But we acknowledge that there are different applications of agents, including asynchronous interactions where a human may not be involved in real-time. In the limitations section of our paper, we have briefly mentioned this constraint (Appendix G. Limitations of the Current User Interface). **We will definitely emphasize this distinction more clearly in the revised manuscript to better position our work.**
>
> > ***W 2(a): When approximation agent and target agent can (often) disagree, it can lead to user confusion and lack of transparency on what is going on on UI.***
>
> Thank you for highlighting this important aspect. **To address this, our user interface design, as described on page 6, incorporates a rescheduling mechanism that carefully manages when and which outputs from both the approximation and target agents are presented to the user.**
>
> Specifically, the user interface only displays the approximation agent’s response based on a confirmed action trajectory. For instance, if the approximation agent quickly generates two sequential actions before the target agent confirms the first one, only the first action generated by the approximation agent will be shown. The system will then wait for the target agent's result on the first step before presenting the next message, which is the target agent’s result on the first step. If the target agent’s result aligns with the approximation agent’s result, the second step computed by the approximation agent will be presented, as it is based on a correct action trajectory.
>
> Consequently, the user will see pairs of (approximation agent’s generated response at step i, target agent’s generated response at step i) sequentially. This design aims to reduce user confusion and enhance transparency, ensuring a clear and coherent interaction experience.
>
> > ***W2 (b): The window for interruption is basically the time for target agent to execute, which is not user-friendly or "user-centric".***
>
> Thank you for raising this critical point. **We would first like to direct your attention to Appendix G. Limitations of the Current User Interface, where we discuss the limitations and potential future directions of the user interface.**
>
> **First of all, we want to emphasize that the current design can be very useful**. For example, in scenarios such as multi-agent discussions, where the target agent may take several minutes to complete a single step. This approach was motivated by our observations of systems like AutoGen, which prompts the user after one round of multi-agent discussion and waits for user input. In many cases, users may lack the patience for multi-round discussions, especially when the action or step under discussion is trivial. This observation motivated us to design a system where users can actively interrupt and interact with the process.
>
> Secondly, a crucial design/research question related here is what and how much information to present to the user: **what information should we display to users, only the final result for each step or the entire thinking process?** In the former case, there would be a longer wait for the step to be presented; in the latter, the volume of text generated could be overwhelming. **This decision also influences whether we expect immediate user interruptions or implement potentially a roll-back mechanism to change what has been planned due to long reading time in user side**. The current system design presents only the final decided step, which reduces the cognitive load on the user. However, a roll-back mechanism would be ideal for scenarios where more information is presented and could also enhance the current presentation mode by offering users more flexibility and the option to be less attentive. Combining immediate user interruptions with a roll-back mechanism could indeed be a valuable future direction. We will consider this enhancement in our ongoing work.

---

> ### Author Response · Authors · 2024-11-18
> **Rebuttal for Reviewer 2 for Weakness 2(c), Question 1 & 2**
>
> > ***W 2(c): I did not fully understand how this mechanism can work when the action execution can be anything of consequence in the user interaction (e.g. sending an email or request for money to split bill on a mobile pay application).***
>
> Thank you for raising this important concern. **We would first like to direct your attention to Appendix G. Spectre**, where we have discussed the issue of speculative execution vulnerabilities, commonly referred to as Spectre.
>
> **Spectre (security vulnerability) is well-documented in the literature (McIlroy et al., 2019; Kocher et al., 2020) and pertains to vulnerabilities involved in speculative execution**, a hardware feature that enhances processor performance by predicting and executing future instructions. Speculative execution and its associated risks have a long history in operating systems.
>
> Given the potential for incorrect and irreversible actions, such as sending emails or financial transactions, **we agree that the current form of Interactive Speculative Planning should be applied cautiously**. Specifically, we recommend constraining its use to non-high-stakes areas where the consequences of incorrect actions are minimal. **In Appendix G, we also outline potential solutions for safer Interactive Speculative Planning, aiming to broaden its applicability while ensuring security and reliability.**
>
> We will continue to explore and address these limitations in our ongoing work.
>
> > ***Q1: Could a theoretical focus on system latency lead to newer issues in the human-AI interaction?***
>
> A theoretical focus on system latency can indeed introduce new considerations and potential issues in human-AI interaction. Below are 2 examples:
>
> (1) the relationship between user experience and the amount of text presented is crucial. If the presented text is very long and dense, decreasing latency may not necessarily improve user experience. Conversely, if the text is concise and well-summarized, reducing latency could ideally enhance user satisfaction by providing timely and digestible information.
>
> (2) designing an interactive system with a theoretically guaranteed latency could be highly beneficial. Such a system would reassure users that, regardless of their interaction style, the time the system spends on a task is guaranteed to be below a certain threshold. This predictability can enhance user trust and satisfaction, knowing that the system will respond within a reliable timeframe.
>
> > ***Q2: Is this approach suitable for all tasks? A more detailed discussion would be useful.***
>
> Thank you for raising this important question. Currently, there are two domains where the present implementation may fall short:
>
> (1) **High-stakes domains**: As mentioned in Appendix G, "Limitations and Future Directions. Spectre," the current implementation is not suitable for high-stakes domains.
>
> (2) **Agentic tasks where the order of steps is not crucial**: These tasks may also be unsuitable, as there could be many false negatives in rejecting the approximation agent's plan if the steps are simply in a different (but not incorrect) order compared to the target agent.
>
> We appreciate your feedback and will move the relevant content from the appendix to the main body of the paper to provide a more detailed discussion on the suitability of this approach for various tasks.
>
> ---
>
> We really appreciate your time and consideration in reviewing our paper. We will really appreciate it if you could re-assess our work, especially if our explanations have clarified any previous ambiguities. We will incorporate further motivation in our revised manuscript.

---

> ### Author Response · Authors · 2024-11-23
> **Further discussion and potential score increase?**
>
> Dear Reviewer y5dF,
>
> We sincerely appreciate the time and effort you've invested in reviewing our paper. Your insightful comments and thoughtful feedback are invaluable to us.
>
> We hope that our responses have addressed your questions and clarified any ambiguities. Please do not hesitate to reach out if you have any further questions!
>
> **We would be truly grateful if you would consider our clarifications and kindly reevaluate your score.**
>
> Thank you once again for your consideration.
>
> Best regards,
> the authors

---

> > ### Comment · Reviewer_y5dF · 2024-11-26
> >
> > Thank you for the detailed rebuttal and that answers many of the concerns  I have now read the comments from other reviewers and also the rebuttal to those comments.
> >
> > The only minor comment to the rebuttal i had was about the statement "there are two domains where the present implementation may fall short: High-stakes domains..". I am glad authors acknowledge that when task may include subtasks that are irreversible, this may not be a good idea. However, in the intro as part of the motivation, the authors state "Particularly in scenarios where complex tasks are delegated to LLM-based agents, often involving high stakes and complex decision-making processes, users may not anxiously wait for the agent to respond all at once, but rather expect the agent to provide timely feedback". This gives the impression that speculative planning is particularly suited for high stakes domain. Anyway, since this is rather minor rephrasing comment , i am happy to raise my score as the rebuttal clarifies the questions i had.

---

> > > ### Author Response · Authors · 2024-11-26
> > > **Thank you for raising the score!**
> > >
> > > Dear reviewer,
> > >
> > > Thank you very much for your appreciation!
> > >
> > > Thank you again for pointing out the phrasing issue and I will re-paraphrase the part.
> > >
> > > Best,
> > > the authors

---

### Official Review · Reviewer_SYAX · 2024-11-05

**Soundness:** 3
**Presentation:** 3
**Contribution:** 3
**Rating:** 8
**Confidence:** 4

**Summary:**

Describes a speculative planning algorithm for LLM agents that assumes an approximate model and a target model. It is assumed that the target model is more capable but slower than the approximate model. Planning is performed by the approximate model until it is deemed to deviate from the target model, at which time the approximate model is corrected. Human interaction is also addressed via a rescheduling algorithm and the ability to interrupt and modify the plan.

**Strengths:**

The problem is relevant, agents are slow and we should try to make them more responsive.

The approach is novel afaik.

The algorithm itself is clearly described.

**Weaknesses:**

The algorithm trades off time with efficiency. Are there ideas to also improve efficiency?

The selection of an approximate model and the target model may be difficult to satisfy, and increase system complexity in practical applications.

**Questions:**

The evaluation could be improved by comparing a smaller and larger model from a particular model family, i.e. 8b vs. 70b llama using the same approach.

Are there more efficient approaches to validating the approximate models plan without comparing directly to the target models plan?

If the user interrupts the plan, do you envision that they edit the agents trajectory directly? or provide feedback and have the agent regenerate the step?

---

> ### Author Response · Authors · 2024-11-18
> **Rebuttal for Reviewer 1 for weakness 1 & 2**
>
> We would like to extend our sincere gratitude for the invaluable time and effort you've dedicated to reviewing our manuscript and for providing us with detailed feedback.
>
> Below are our reply to the two main concerns you raised:
>
> ---
>
> > ***W1: The algorithm trades off time with efficiency. Are there ideas to also improve efficiency?***
>
> Thank you for raising this important question. There are several notable approaches in the literature that address cost reduction in agent planning:
>
> **EcoAssistant**: The paper titled "EcoAssistant: Using LLM Assistant More Affordably and Accurately" employs a model cascade to reduce planning costs by initially using a smaller, more efficient model. While this approach saves cost, it does so at the expense of increased latency.
>
> **System-1.x**: The paper "System-1.x: Learning to Balance Fast and Slow Planning with Language Models" fine-tunes a controller, a System-1 Planner, and a System-2 Planner based on a single LLM. This system uses the controller to decide whether to use System-1 or System-2 for planning specific steps, thereby reducing costs. However, this method requires model training on specific tasks.
>
> **Online Speculative Decoding**: The paper is about speculative decoding but I think the idea can be adopted to speculative planning. Online Speculative Decoding proposes tuning the approximation LLM based on feedback from the larger target LLM. This approach enhances the accuracy of the approximation model and reduces latency through online learning. A similar idea could be applied in our context to improve accuracy of the approximation agent and thus reducing the cost (fewer wasted processes) as well as latency.
>
> **Currently, I do not have any other brand-new ideas for reducing both latency and cost beyond what these models have discussed. And in general, I think model training may be a necessary step if we want to reduce both latency and cost.** However, this is at the top of my to-do/to-think list, and I am actively exploring potential solutions.
>
> Thank you for your insightful feedback. We will continue to investigate ways to optimize both cost and latency in our future work.
>
> ---
> > ***W2: The selection of an approximate model and the target model may be difficult to satisfy, and increase system complexity in practical applications.***
>
> Thank you for raising this important consideration. The choice of the approximation agent is indeed crucial for the system's effectiveness.
>
> **Speculative planning is compatible with a very broad range of approximation agents for a given target agent** (unlike speculative decoding, where the approximation models must be from the same group to ensure they share the same vocabulary for sampling): For a given target agent, we can select agents with the same backbone LLM but different prompting styles or available tools. Alternatively, we can choose agents with weaker backbone LLMs using the same or different prompting methods, or employ more complex prompting methods to match performance. Additionally, we can opt for multi-agent systems with various ensembles of agents.
>
> **This wide range of possibilities is more positive than the downside**. This wide range of possibilities means we have extensive opportunities to optimize the system and find the most suitable agent or multi-agent configuration. While selecting the "right" approximation agent from such a broad spectrum may seem challenging, it also presents a significant advantage: greater flexibility and a wider range of possibilities for optimization. This flexibility allows us to tailor the system more precisely to various needs and scenarios.
> Comparing with the target agent’s plan on some agentic benchmark is definitely the most reliable method to determine which approximation agent to use.
>
> **Regarding the issue of increasing system complexity, this can be mitigated through careful design and packaging.** If well-designed and packaged, such a system should be easily leveraged, similar to speculative decoding, which has been integrated into many LLM serving platforms and packages such as vLLM. Currently, the system can be used off-the-shelf, thus it does not significantly enhance complexity in applications from either the application development side or the user side.

---

> ### Author Response · Authors · 2024-11-18
> **Rebuttal for Reviewer 1 for question 1, 2 & 3**
>
> > ***Q1: The evaluation could be improved by comparing a smaller and larger model from a particular model family, i.e. 8b vs. 70b llama using the same approach.***
>
> Thank you for the suggestion, and it is indeed an interesting design. **But notice that Setting 4 is exactly using the setting you are mentioning, where we utilize GPT-3.5 and GPT-4 with direct generation.** To enrich experiments for this design, here we present the extra experiments (on 50 random datapoints for now) of OpenAGI **using the Llama-3.1 models** with FP8 quantization with Together AI API. The generation strategies are CoT and ReAct.
>
> For each table, we present normal agent planning with 70b model, speculative planning with 8b + 70b model, normal agent planning with 405b model, speculative planning with 70b + 405b model, normal agent planning with 405b model, and speculative planning with 8b + 405b model.
>
> **Result using CoT prompting strategy**
>
> | settings | 70b   | 8b + 70b | 405b   | 70b + 405b | 405b   | 8b + 405b |     |
> | -------- | ----- | -------- | ------ | ---------- | ------ | --------- | --- |
> | TT       | 37.68 | 31.33    | 32.030 | 26.25      | 32.030 | 32.47     |     |
> | ST       | 3.85  | 3.05     | 5.71   | 5.08       | 5.71   | 5.59      |     |
> | TO       | 1880  | 2684.67  | 926.5  | 1662.17    | 926.5  | 1328.19   |     |
> | SO       | 182.4 | 331.12   | 162.82 | 339.08     | 162.82 | 196.0     |     |
> | MC       | 1     | 4.00     | 1      | 3.91       | 1      | 4.52      |     |
> | cost     | 0.007 | 0.011    | 0.014  | 0.018      | 0.014  | 0.021     |     |
>
> **Result using ReAct prompting strategy**
>
> | settings | 70b     | 8b + 70b | 405b    | 70b + 405b | 405b    | 8b + 405b |     |
> | -------- | ------- | -------- | ------- | ---------- | ------- | --------- | --- |
> | TT       | 46.03   | 33.12    | 49.34   | 41.08      | 49.34   | 45.49     |     |
> | ST       | 5.98    | 3.88     | 7.693   | 5.87       | 7.693   | 7.134     |     |
> | TO       | 1074.57 | 1511.33  | 1343.19 | 1840.84    | 1343.19 | 1690.62   |     |
> | SO       | 149.71  | 177.09   | 195.59  | 284.18     | 195.59  | 241.0     |     |
> | MC       | 1       | 4.05     | 1       | 4.10       | 1       | 4.67      |     |
> | cost     | 0.013   | 0.018    | 0.028   | 0.038      | 0.028   | 0.033     |     |
>
> > ***Q 2: Are there more efficient approaches to validating the approximate models plan without comparing directly to the target models plan?***
>
> Thank you for raising this insightful question. If I understand correctly, you are asking how to efficiently evaluate whether the approximation agent's planned step is correct without directly comparing it to the target agent's plan.
>
> Currently, similar to speculative decoding, our approach relies on the target agent for verification, which means we have to wait until the target agent completes the corresponding step. However, there are other potential methods to consider. For instance, we could use an **external evaluator**, which could be the target agent itself if it can perform the evaluation more quickly than computing the step. Alternatively, we could **run the proposed step in a simulated environment**, assuming this process is faster than waiting for the target agent to finish computing.
>
> **While these methods could improve efficiency, they may not guarantee that the final output of such speculative planning system, using both approximation and target agents, would be equivalent to that of normal agent planning using only the target agent**. Currently, by comparing with the target agent's plan, we can ensure output equivalence. Other methods can certainly be explored for better efficiency, but at present, I cannot think of a validation method that both guarantees output equivalence and is faster than comparing with the target model's plan. (Notice that it is not necessarily a bad thing not being able to maintain output equivalence, as it could potentially increase the target agent performance )
>
> We appreciate your feedback and will continue to investigate more efficient validation approaches in our future work.
>
> > ***Q3: If the user interrupts the plan, do you envision that they edit the agents trajectory directly? or provide feedback and have the agent regenerate the step?***
>
> Thank you for raising this important question. **Currently, we envision users editing the agent trajectories directly**. While we have implemented an additional feature to accept user feedback, the latency can vary significantly based on the nature of the user input and how the model interprets this feedback. Given that the current focus of our paper is on developing a system that decreases latency with a strict upper bound, we have not included experimental results for the feedback-taking scenario. We appreciate your insight and will consider exploring this aspect in future work to provide a more comprehensive understanding of user interaction dynamics.
>
> ----
>
> We hope our answers clear your questions. Thank you again!

---

### Author Response · Authors · 2024-12-04
**Summary of Rebuttal**

Dear reviewers, AC, and SAC,

We sincerely thank the reviewers for dedicating their time to our paper and providing such insightful comments. We have summarized the main aspects of the discussion during the rebuttal period into 6 aspects presented below:

> ***1. How to select an appropriate approximation agent (reviewer SYAX, SCq5)?***

Given user preferences or requirements on acceptable ranges of cost increase and time reduction, we can compute all possible configurations (including accuracy and speed) of the approximation agent for each value of k that meet these criteria. The detailed computation and selection process can be found in our response https://openreview.net/forum?id=BwR8t91yqh&noteId=5v5YExOc78. Discussion over advantages of the wide range of choices of approximation agent configurations can be found in https://openreview.net/forum?id=BwR8t91yqh&noteId=6IZ7SdHSwz.

> ***2. How do we choose k in the system (reviewer SCq5)?***

Similarly, based on preferences or requirements for acceptable cost increases and time reduction proportions, we can determine all possible values of k for each configuration of the approximation agent that comply with these requirements. The computation and selection process is detailed in the same response in https://openreview.net/forum?id=BwR8t91yqh&noteId=5v5YExOc78

> ***3. How much cost is required for the system and possible ideas to also reduce cost in the system (reviewer SYAX, SCq5)?***

We have added a cost increase analysis to the revised paper as requested by Reviewer SCq5. Potential ideas to reduce costs, such as model cascading and offline or online model training for the approximation agent, are discussed in our response in https://openreview.net/forum?id=BwR8t91yqh&noteId=6IZ7SdHSwz

> ***4. Additional experiments using a smaller and larger model from a particular model family with the same prompting approach (reviewer SYAX)?***

Experimental results using Llama 3.1 models of varying sizes (8B, 70B, 405B) with Chain-of-Thought and ReAct generation are presented in our response Q1 in https://openreview.net/forum?id=BwR8t91yqh&noteId=TVIDUMdXDi. All additional experiments show noticeable time reductions in agent planning, supporting our main motivation.

> ***5. How does the design of the UI design/scheduling mechanism enables more user-friendly user interface (reviewer y5dF, SCq5)?***

Our UI design encapsulates the complex concurrent processes of speculative planning into a sequential and easily understandable format. This approach allows users to perceive the computation latency associated with the target agent and how computation time is saved by the approximation agent. It also represents a technical contribution in building the first system that allows synchronous real-time human-AI interaction, which is a scenario where latency matters the most. Relevant answers can be found in https://openreview.net/forum?id=BwR8t91yqh&noteId=hYa22oPSPP and  https://openreview.net/forum?id=BwR8t91yqh&noteId=GruMFkfDbG

> ***6. The range of applicable domains of the acceleration method (reviewer y5dF)***

As mentioned in Appendix G, "Limitations and Future Directions. Spectre," the current implementation is not suitable for high-stakes domains. We also emphasized it in Q2 of https://openreview.net/forum?id=BwR8t91yqh&noteId=33Tu4b7I4V.

---

### Meta-Review · Area_Chair_5quc · 2024-12-19

**Metareview:**

The paper focuses on proposing a new method for designing an agent system as well as a user interface together - placing user interactions in the center. As agents begin to automate more and more tasks, such methods will become very relevant. All reviewers are also unanimous in their agreement about the merits of the paper.

**Additional Comments On Reviewer Discussion:**

There was reasonable discussion during the rebuttal phase and a good faith attempt to answer all the reviewers' concerns by the authors. SCq5 increased their score and many of the other concerns by others were seemingly addressed.

---

### Decision · Program_Chairs · 2025-01-22

Accept (Poster)